**Photosynthetic electron, carbon and oxygen fluxes within a mosaic of Fe limitation in the California Current Upwelling System**

Yayla Sezginer[1], Kate Schuler[1], Emily Speciale[2], Adrian Marchetti[2], Claire Till[3], Ralph Till[3], Philippe Tortell[1,4]

[1]Department of Earth, Oceans and Atmospheric Sciences, University of British Columbia, Vancouver, BC, Canada
[2]Earth Marine and Environmental Sciences, University of North Carolina at Chapel Hill, Chapel Hill,
NC, USA
[3]Department of Chemistry, California State Polytechnic University, Humboldt, Arcata, CA, USA
[4]Department of Botany, University of British Columbia, Vancouver, BC, Canada

*Correspondence to*: Yayla Sezginer (ysezginer@eoas.ubc.ca)

**Abstract.** We compare primary productivity estimates based on different photosynthetic 'currencies'
(electrons, $O_2$ and carbon) measured in the dynamic coastal upwelling waters of the California Current.
Fast Repetition Rate Fluorometry and $O_2/N_2'$ measurements were used to collect high-resolution
underway estimates of photosynthetic electron transport rates and net community productivity,
respectively, alongside on-station $^{14}C$ uptake experiments to measure gross carbon fixation rates. Our
survey captured two upwelling filaments at Cape Blanco and Cape Mendocino with distinct
biogeochemical signatures and iron availabilities, enabling us to examine photosynthetic processes
along a natural iron gradient. Significant differences in photo-physiology, cell sizes, $Si:NO_3^-$ draw-down
ratios, and molecular markers of Fe-stress indicated that phytoplankton assemblages near Cape
Mendocino were Fe-stressed, while those near Cape Blanco were Fe-replete. Upwelling of $O_2$-poor
deep water to the surface complicated $O_2$-based net community productivity estimates, but we were able
to correct for these vertical mixing effects using continuous $[N_2O]$ surface measurements and depth-
profiles of $\frac{\partial[O_2]}{\partial[N_2O]}$. Vertical mixing corrections were strongly correlated to sea surface temperature,
which serves as an $N_2O$-independent proxy for upwelling. All three productivity estimates reflected
trends in Fe-stress physiology, indicating greater productivity near Cape Blanco compared to Cape
Mendocino. For all phytoplankton assemblages, carbon fixation varied as a hyperbolic function of
photosynthetic electron transport rates, but the derived parameters of this relationship were variable and
significantly correlated with physiological indicators of Fe-stress ($\sigma_{PSII}$, $F_V/F_M$, $Si:NO_3^-$ and diatom-
specific PSI gene expression), suggesting that iron availability influenced the coupling between
photosynthetic electron transport and carbon fixation. Net community productivity showed strong
coherence with daily-integrated photosynthetic electron transport rates across the entire cruise track,
with no apparent relationship with Fe-stress. This result suggests that fluorescence-based estimates of

gross photochemistry are still a good indicator for bulk primary productivity, even if Fe-limitation influences the stoichiometric relationship between different productivity currencies.

## 1 Introduction

Along the eastern boundaries of ocean basins, coastal upwelling delivers nutrient-rich deep water to the euphotic zone, sustaining high phytoplankton growth rates and primary productivity (Bograd et al., 2023). Despite representing less than 1% of the surface ocean, these productive upwelling ecosystems, support ~20% of global fishery catches (Pauly and Christensen, 1995), and play a disproportionate role in ocean carbon uptake through the 'biological carbon pump'(Mathis et al., 2024). Quantifying rates of

primary productivity (PP) within eastern boundary currents is thus vital for accurate carbon budgeting and fishery yield predictions (Marshak and Link, 2021), yet this remains challenging due to the highly dynamic nature of these systems.

The California Current system (CCS) is one of the best studied eastern boundary currents, extending

from British Columbia, Canada, to Baja California, Mexico. Upwelling in the CCS occurs during spring and summer when northerly winds drive Ekman transport of surface water offshore. Within the upwelling season, short-term changes in windspeed and direction can dampen or reverse upwelling signals on the scale of hours to days, while complex coastline geometry directs wind flow, creating upwelling hotspots in the lee of capes (Castelao and Luo, 2018). Underlying bathymetric features and

deep-water composition further influence the nature of upwelling filaments and the availability of macro and micro nutrients. In regions with shallow and wide continental shelves, sediment deposition provides a primary source Fe and other micronutrients (Deutsch et al., 2021). In contrast, waters overlying steep narrow shelves retain less Fe, and PP in these regions can be limited by Fe availability despite the presence of upwelling conditions (Biller et al., 2013). The resulting 'mosaic of Fe limitation'

influences the distribution of phytoplankton biomass and productivity across the CCS (Till et al., 2019; Hutchins et al., 1998). Resolving ecosystem responses to such a heterogenous environment requires high resolution measurements.

Traditionally, primary productivity has been measured using discrete bottle incubations where the net

change in dissolved $O_2$ or particulate organic carbon is measured over time. Shorter incubations approximate gross primary productivity (GPP), whereas longer incubations allow time for respiration to act upon tracers, yielding estimates somewhere between GPP and net PP (NPP = GPP – respiration). Although bottle incubations are still widely used to directly observe carbon fixation rates, the resulting measurements can be ambiguous in terms of GPP vs. NPP, while also providing low sampling

resolution and posing potential containment artefacts (Banse, 2002). To avoid these challenges, a number of high through-put PP proxies have been developed based on advances in dissolved gas measurements, bio-optical techniques, and satellite-based ocean color observations (IOCCG, 2022). These diverse PP methodologies target different photosynthetic processes, from subcellular light absorption to ecosystem scale carbon export.

At the smallest spatial and temporal scales, Photosystem II (PSII) electron transport rates ($ETR_{PSII}$) quantify light absorption and conversion to chemical energy for a variety of metabolic activities including carbon fixation. Measurements of $ETR_{PSII}$ can be obtained from active chlorophyll fluorescence techniques, which exploit the inverse relationship between PSII fluorescence and

photochemical yields to enable non-invasive and high-frequency measurement from underway seawater lines (Kranz et al., 2020; Sezginer et al., 2023) or autonomous platforms (e.g. Carvalho *et al.*, 2020). Downstream of $ETR_{PSII}$, carbon fixation can be directly measured using bottle incubations, or approximated from empirical algorithms relating NPP to remotely sensed Chlorophyll (Chl) concentration, sea surface temperature (SST), and photosynthetically available radiance (PAR)

(Behrenfeld et al., 2005; Saba et al., 2011; Behrenfeld and Falkowski, 1997). Finally, net community productivity (NCP), represents the difference between GPP and community-wide respiration and can be equated to carbon export out of the mixed layer. This term can be derived from measurements of biological oxygen saturation, $\Delta O_2/Ar$, using Ar-normalization of $O_2$ to correct for physical influences on gas disequilibria (e.g. temperature or salinity changes or bubble injection) to isolate the biological signal

(Cassar et al., 2009; Craig and Hayward, 1987). Assuming steady-state conditions, $O_2$ fluxes from the mixed layer represent a balance between net biological production and sea-air exchange, allowing the calculation of NCP from sea-air flux estimates. Steady state assumptions are violated in upwelling systems, such as the CCS, where $O_2$ fluxes in the mixed layer are also affected by vertical mixing. However, such vertical mixing effects can be corrected using $N_2O$ as a tracer of $O_2$-depleted deep water,

given the strong stoichiometric relationship between $N_2O$ and $O_2$ ratios in sub-surface waters (Cassar et al., 2014).

    The various measurement techniques described above yield PP estimates in a number of different 'currencies', i.e. carbon, oxygen and electrons, each with different integration time scales (seconds to

weeks). In theory, combining PP quantification approaches can fill data gaps (e.g. cloud or ice interference with satellite data, or missing years in time series operations), and provide deeper understanding of ocean metabolism. In practice, incorporating alternative measurement approaches requires understanding of conversion rates between the various PP currencies. Predicting these conversion factors is challenging, as they vary in response to environmental conditions and

phytoplankton taxonomy and physiology (Hughes et al., 2021; Hughes et al., 2018a; Schuback et al., 2017; Halsey and Jones, 2015). For example, the $ETR_{PSII}$ : GPP ratio often exceeds the theoretical stoichiometry of 4 (Halsey and Jones, 2015), implying that redox potential generated at PSII is used for functions other than carbon fixation, such as nitrogen uptake or cyclic electron transport. Similarly, differences between $ETR_{PSII}$ and NCP represent combined $O_2$ consumption pathways, including

community-wide respiration, chloro- and photorespiration and pseudo cyclic electron transport. Investigating the drivers of decoupling between currencies can thus improve conversion rate estimates, and also provide insights into energy transfer efficiencies between different components of the photosynthetic process.

Here we present simultaneous PP measurements in the CCS collected using high resolution, underway sampling techniques along a cruise track between Newport, OR and San Francisco, CA. Underway

measurements of $ETR_{PSII}$ and NCP conducted during the May-June upwelling season of 2023 were complemented with parallel measurements of $ETR_{PSII}$ and $^{14}C$-GPP collected at oceanographic stations. Our results demonstrate fine-scale spatial patterns in GPP and NCP associated with variability in localized upwelling filaments, with particularly notable differences observed across gradients of dissolved iron concentrations resulting from variable coastal bathymetry. These results enable us to examine variability in productivity currency conversion factors across natural Fe availability gradients within the California Current System, with potential application to other complex coastal waters.

## 2 Methods and Materials

### 2.1 Sampling Sites

Measurements were collected along the Oregon and Northern California coast during the Phytoplankton UPwelling Cycle (PUPCYCLE II) expedition, onboard the *R/V Sally Ride* from May 27 – June 11, 2023. A key objective of the program was to examine the evolution of phytoplankton blooms in recently upwelled waters, and we specifically targeted two upwelling plumes off Cape Blanco and Cape Mendocino, which were identified by low sea surface temperature (SST < 12°C; Figure 1). Along the cruise track, temperature, salinity, and chlorophyll fluorescence were monitored by the ship's underway system, supplied by a seawater supply line with a nominal intake depth of approximately 5m. Nitrate concentrations were measured continuously with a Seabird SUNA sensor calibrated against $NaNO_3$ standards immediately prior to the cruise. In addition to standard oceanographic variables, the seawater supply line was also used for continuous underway measurements of phytoplankton photo-physiology and $ETR_{PSII}$ using a Fast Repetition Rate fluorometer (FRRF; Soliense Inc.), and NCP using a custom-built Pressure In-Situ Gas Instrument (PIGI), as described below.

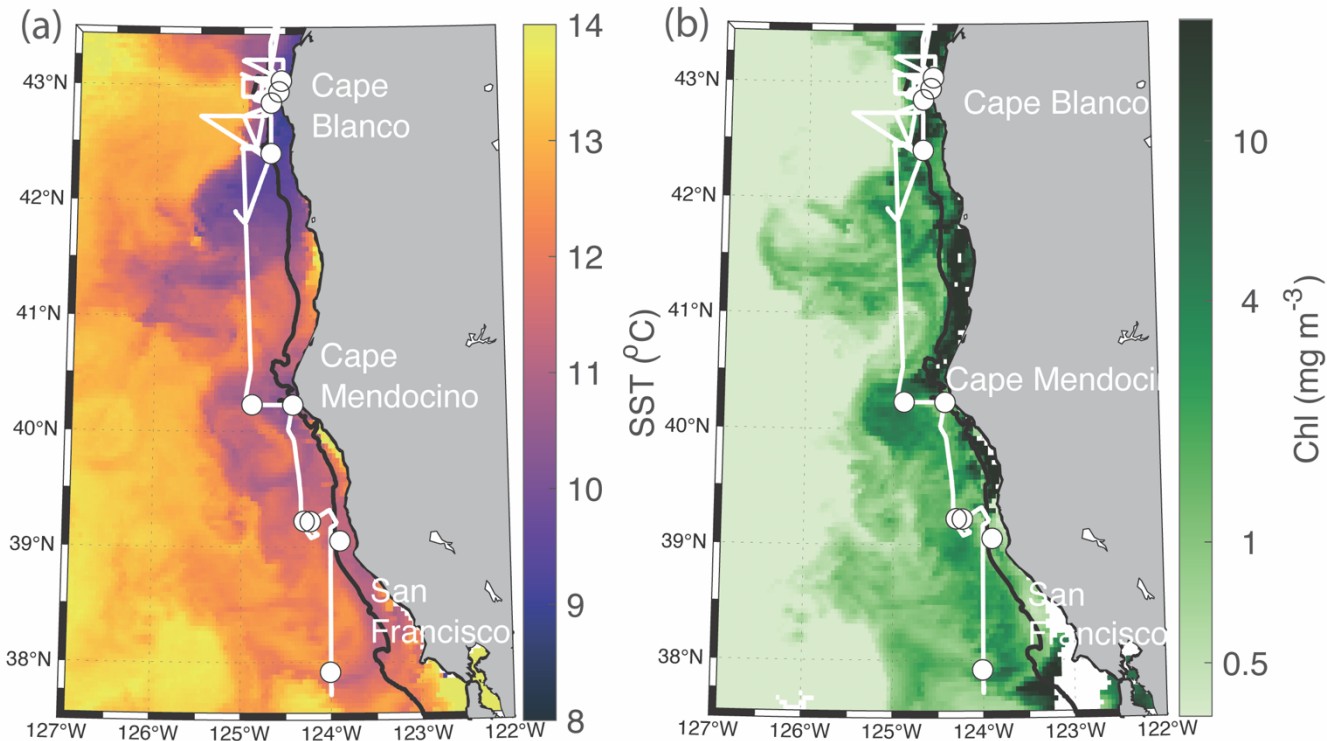

**Figure 1**. **Study site map.** Location of the cruise track (white line) and discrete sampling stations (white circles) during the PUPCYCLE expedition. The black bathymetric contour line represents the 200 m isobath. The study region and surrounding waters are colored by **a.** daily NASA Aqua Modis Level 3 satellite retrievals of sea surface temperature (SST) averaged over the cruise period (May 27 – June 11, 2024) **b.** NASA Aqua Modis Level 3 satellite retrievals of Chlorophyll *a* (Chl) averaged over the cruise period.

Daily CTD and rosette casts were conducted one hour before local sunrise, and samples were collected at four depths targeting 1%, 10%, 22%, and 46% of surface solar irradiance levels. Photosynthesis-Irradiance (PI) curves (measured with Fast Repetition Rate fluorometry– see below), chlorophyll *a* (Chl) and nutrient concentrations ($NO_3^-$, Si, $PO_4$, Fe) were measured at all four depths. Samples collected from 1% and 46% irradiance levels were also incubated for $^{14}C$-based PI curves (see sect. 2.5), with sub-samples collected for High Performance Liquid Chromatography (HPLC)-based analysis of phytoplankton pigments.

## 2.2 Fast Repetition Rate Fluorometry

A bench-top Soliense Inc. Fast Repetition Rate Fluorometer (FRRF) was configured for underway collection of chlorophyll *a* fluorescence transients following Sezginer *et al.*, (2021). Samples were introduced to the measurement cuvette by an integrated peristaltic pump. The pump was used to flush the measurement cuvette for 2.5 minutes before isolating a sample for analysis. Following a one-minute

dark period to relax short lived non-photochemical quenching, five single-turnover fluorescence
transients were collected from each sample in the dark. Each single-turnover transient included a
sequence of 100 sub-saturating excitation light pulses of 1.5 $\mu$s duration, and a 1 $\mu$s interval between
pulses. The excitation phase is designed to induce photochemistry and gradually reduce the pool of
primary electron acceptor molecules, $Q_a$ (Kolber et al., 1998). As the $Q_a$ pool is reduced, further
electron exchange between PSII and $Q_a$ is prevented, closing the photochemistry pathway and causing a
concurrent increase in PSII fluorescence yields. The excitation phase is followed by a relaxation
sequence consisting of 127 light pulses with an initial 20 $\mu$s interval. During the relaxation phase, the
interval between light pulses increases exponentially, enabling $Q_a$ reoxidation between pulses to
gradually reopen the photochemistry pathway, such that fluorescence yields return to their basal levels.
The biophysical model described by Kolber et al. (1998) was fit to the resulting fluorescence transient
to derive the maximum quantum yield of PSII ($F_V/F_M$), functional absorption area of PSII ($\sigma_{PSII}$), and
turnover rate of the primary electron acceptor $Q_a$ ($\tau_{Qa}$).

## 2.3 Electron Transport Rates, ETR$_{PSII}$

Continuous FRRF sampling was interrupted every 12 samples (~3hrs) to conduct a Photosynthesis-
Irradiance (PI) curve characterizing light-dependent changes in photo-physiology and photochemistry.
Across the entire cruise, we collected 91 PI curves. Each PI curve was initiated with a fresh sample, and
consisted of 11 light levels, increasing from $0 - 850$ $\mu$mol photons m$^{-2}$ s$^{-1}$. Light was supplied evenly by
the five actinic LEDs within the FRRF (445, 470, 505, 530, and 590 nm) calibrated against a handheld
WALZ ULM-500 PAR meter prior to deployment. At each light level, ETR$_{PSII}$ was calculated following
Suggett and Moore (2010):

$$ETR_{PSII} = PAR * \sigma'_{PSII} * \frac{F'_q}{F'_v} * 6.033 * 10^{-3}$$

1.

In this formulation, the rate of photon delivery to the pool of PSII reaction centers (RCII) is derived as
the product of photosynthetically active radiation (PAR; units $\mu$mol photons m$^{-2}$ s$^{-1}$) supplied by the
LED lamps, and the wavelength-specific functional absorption area of RCII ($\sigma'_{PSII}$; units of Å$^2$ PSII$^{-1}$).
The conversion efficiency from light to photochemical energy depends on the fraction of open RCII,
measured as the dimensionless ratio between the fluorescence amplitude measured under actinic light
when photochemistry is active ($F'_q = F'_m - F'$) and that measured under actinic light if all RCII were
oxidized, ($F'_v = F'_m - F'_o$). In practice, measuring the minimum fluorescence when the RCII pool is
completely open ($F'_o$) is challenging, as actinic light always drives some degree of photochemistry and
reduction of RCII. Alternatively, $F'_o$ can be derived as $F_o/(F_V/F_M + F_o/F'_m)$, following Oxborough $et$ $al.$,
(2012).

The prime notation (') refers to FRRF parameters derived under actinic light. The constant $6.033 * 10^{-3}$
converts PAR to units of quanta m$^{-2}$ s$^{-1}$, and $\sigma'_{PSII}$ to units of m$^2$ RCII$^{-1}$, yielding ETR$_{PSII}$ units of quanta

s$^{-1}$ RCII$^{-1}$. For a complete description of FRRF derived parameters, see Schuback *et al.* (2021) and Tortell et al. (2023).

For each light curve measured, ETR$_{PSII}$ was plotted against PAR and fit with the photosynthesis-irradiance model of Platt et al. (1980):


$$ETR_{PSII} = P_s \left(1 - e^{-\frac{\alpha PAR}{P_s}}\right) e^{-\frac{\beta PAR}{P_s}} \qquad\qquad 2.$$

During the initial light-limiting part of the curve, $ETR_{PSII}$ increases linearly with PAR with a slope of $\alpha$. As PAR increases to saturating levels, $ETR_{PSII}$ stabilizes at maximum levels, P$_{max}$. The light saturation index, E$_k$ is derived as P$_{max}$/$\alpha$. When phytoplankton are affected by photoinhibition, $\beta$
describes the decrease in $ETR_{PSII}$ at high light levels (i.e. >> E$_k$). In the absence of photoinhibition, P$_s$ = P$_{max}$. When photo-inhibition is present ($\beta$ > 0), P$_s$ represents the theoretical maximum potential $ETR_{PSII}$. When $\beta$ > 0, P$_{max}$ is derived as:

$$P_{max} = P_s \left(\frac{\alpha}{\alpha + \beta}\right) \left(\frac{\beta}{\alpha + \beta}\right)^{\left(\frac{\beta}{\alpha}\right)} \qquad\qquad 3.$$

To evaluate $ETR_{PSII}$ over the cruise track, we linearly interpolated derived values of $\alpha, \beta,$ and P$_s$ to the
sampling resolution of continuous PAR measurements from the ship's meteorological tower (Biospherical Inst. QSR-240P). The mean light intensity in the mixed layer (PAR$_{in\ situ}$) was estimated by accounting for light attenuation with depth, quantified by the diffuse attenuation coefficient (K$_d$ = ln(PAR$_0$/PAR$_{mld}$)/(mld - 0) ), where PAR$_0$ is PAR measured at the surface and PAR$_{mld}$ is PAR measured at the mixed layer depth (mld) (Domingues and Barbosa, 2023). For each CTD cast (n = 28), mld was
determined using a density difference criterion of 0.125 kg m$^{-3}$, and PAR$_{mld}$ was measured with a Biospherical QSP-200 PAR sensor mounted to the CTD rosette. Both K$_d$ and mld were linearly interpolated to the resolution of continuous PAR measurements. Finally, in-situ PAR was estimated following Domingues and Barbosa (2023) as,

$$PAR_{in\ situ} = PAR_0 (1 - e^{-K_d * MLD})(K_d * mld)^{-1} \qquad\qquad 4.$$


To compare ETR$_{PSII}$ with NCP (Sect. 2.6), we converted ETR$_{PSII}$ from e$^-$ RCII$^{-1}$ s$^{-1}$ to volumetric units of mmol O$_2$ m$^{-2}$ d$^{-1}$. This conversion requires an estimate of the chlorophyll content of RCII, which is known to vary significantly across phytoplankton in response to taxonomic and environmental influences (Greene et al., 1992; Murphy et al., 2017; Aardema et al., 2024). Following previous authors
(Schuback et al., 2015; Kolber and Falkowski, 1993), we assumed a possible range of Chl to RCII ratios of 400 to 700, yielding upper and lower bounds of Chl-normalized ETR$_{PSII}$.

$$GP = ETR_{PSII} * 86400 * (Chl:RCII)^{-1} * Chl * mld * \frac{1}{4}$$ 5.

To obtain volumetric units, Chl-normalized ETR was multiplied by mixed layer Chl concentrations (mmol Chl m$^{-3}$ * mld), assuming homogenous [Chl] throughout the mixed layer. Multiplying by 86400
converts from s$^{-1}$ to d$^{-1}$. Given that four charge separation events are required per $O_2$ evolved, ETR was divided by 4 for final gross photochemistry (GP) estimates in terms of mmol $O_2$ m$^{-2}$ d$^{-1}$.

## 2.4 Non-Photochemical Quenching, NPQ

Under excess irradiance, light supply to PSII outpaces maximum downstream electron transport rates, creating the potential for dangerous reactive oxygen species (ROS) to accumulate (Müller et al., 2001).
To mitigate excess excitation, photoautotrophs have evolved a number of photoprotective mechanisms, including non-photochemical quenching (NPQ), which dissipates excitation absorbed by PSII as heat, thereby reducing PSII photochemical and fluorescence yields. Previously, NPQ has been quantified from FRRF data as Stern-Volmer quenching, defined as the relative decrease in PSII fluorescence in response to light exposure: $NPQ_{SV} = (F_m - F'_m)/F'_m$. However, this formulation does not account for
longer-lived NPQ mechanisms that may still be active during dark measurements following recent high light exposure. To overcome this limitation, we used the normalized Stern-Volmer parameter (NPQ$_{NSV}$), calculated as $F'_o/F'_v$ (McKew et al., 2013). For each PI curve measured, NPQ$_{NSV}$ was plotted against PAR and fit with a single component exponential curve. Out of 91 curve fits, 95% had $R^2 > 0.90$ and 87% had $R^2 > 0.95$. In-situ NPQ$_{NSV}$ was then estimated by mapping the resulting curve fit onto in-situ
PAR values.

Similarly, for each PI curve, the fraction of closed RCII, $F'_q/F'_v$, was plotted as a function of PAR and fit with a single component exponential curve, with 92% of curves having an $R^2 > 0.90$. Following the same procedure used to evaluate in-situ NPQ$_{NSV}$, in-situ $F'_q/F'_v$ was approximated by mapping the
resulting curve fit onto in-situ PAR values.

## 2.5 $^{14}$C-uptake experiments

During daily station sampling, 200 mL were collected from 1% and 46% light level depths (approximately 50 and 10 meters, respectively) into acid washed 250 mL bottles for $^{14}$C incubations. Samples were immediately spiked with 150 $\mu$Ci of H$^{14}$CO$_3$ (Perkin Elmer), and inverted to homogenize
the contents of the bottles. The homogenized media was then aliquoted into 20 mL borosilicate scintillation vials, which were incubated over 3 hours in a custom-built photosynthetron at 7 light levels from $0 - 650$ $\mu$mol photons m$^{-2}$ s$^{-1}$. At the end of the incubation, the entire content of the vials was filtered onto 25mm GF/F filters with a nominal pore size of 0.7 $\mu$m. Filters were fumed with 10% HCl for 24 hours to remove any inorganic carbon prior measuring activity on filters with an on-board
scintillation counter (Beckman LS 6500). Immediately after spiking samples, three vials were filtered

for triplicate time zero measurements. Three 100 $\mu$L aliquots were also taken from the initial 200 mL sample and treated with 100 $\mu$L of 3M NaOH to measure total $^{14}$C counts. Disintegrations per minute were converted into hourly C fixation rates according to Knap *et al.*, (1996).

## 2.6 Net Community Productivity (NCP)

We measured NCP based on mixed layer concentrations of $O_2$ and $N_2$ obtained from the Pressure of In-Situ Gas Instrument (PIGI), following Izett and Tortell (2021) and Izett *et al.* (2021). This method estimates NCP from the biological oxygen saturation anomaly, $\Delta O_2/N_2'$, using $N_2'$ as an analog for Argon (Ar) to correct for physical effects on $O_2$ saturation. In this method, net community productivity is equated to the sea-air flux of $O_2$ as determined by the biological saturation anomaly ($\Delta O2/N2'$)
scaled by the [$O_2$] in equilibrium with the atmosphere ([$O_2$]$_{sat}$), and the $O_2$ piston velocity ($k_{O_2}$).

$$\text{NCP} = k_{O_2} * \Delta \frac{O2}{N_2'} * \ [O_2]_{sat} \qquad\qquad 6.$$

The PIGI enables cost effective measurements of $\Delta O_2/ N_2'$ using an oxygen optode and a gas tension device rather than a mass spectrometer, which is more commonly used to measure $\Delta O_2/Ar$ (Izett and
Tortell, 2021). In this method, $N_2'$ is derived as an approximation of Ar, using model calculations that quantify differences between Ar and $N_2$ concentrations due primarily to solubility changes and bubble processes. A full description of the 1D model applied to estimate $N_2'$ is available in Izett and Tortell (2021). The model uses ancillary data, including windspeed, mixed layer depth, temperature, salinity, and sea level pressure, to estimate changes in mixed layer Ar and $N_2$ concentrations over one residence
time period prior to sampling. We applied a residence time of 14 days for this region where mixed layer gas residence times are strongly influenced by the timescales of upwelling events (Austin and Barth, 2002). Ancillary datasets required for $N_2'$ calculations were obtained from a combination of satellite observations and model products, and are compiled in the Supplement S1, alongside descriptions of each data source. The 1D model calculations and code are available at
https://github.com/rizett/O2N2_NCP_toolbox with example calculations.

Additional corrections to NCP estimates were made to account for vertical mixing fluxes, which transport low $O_2$ water to the surface (Izett et al., 2018). Previous studies have omitted $\Delta O_2/Ar$ data collected in known upwelling areas, where the assumption of limited vertical mixing fluxes on $\Delta O_2/Ar$
variability is violated (e.g. Stanley *et al.*, 2010). To address this limitation, Cassar et al. (2014) developed an approach to use surface measurements of $N_2O$ to quantify vertical transport of low $O_2$ waters. In marine environments, there is a strong stoichiometric relationship between apparent oxygen utilization and $N_2O$, which is produced as a by-product of subsurface oxygen-consuming N remineralization pathways (Elkins *et al.*, 1978). These $N_2O$ production pathways are thought to be
photo-inhibited within the euphotic zone (Olson, 1981; Horrigan et al., 1981), so that excess $N_2O$ concentrations in the mixed layer serve as a tracer for vertical influxes of $O_2$-depleted subsurface water.

We thus used the approach of Cassar et al. (2014) and Izett *et al.* (2018), to correct for vertical mixing following Eq. 7.

$$\text{NCP} = k_{O_2} * \left( \frac{\Delta O_2}{N_2{'}} * [O_2]_{\text{sat}} - \frac{k_{N_2O}}{k_{O_2}} * \frac{\partial [O_2]^B}{\partial [N_2O]^B} * [N_2O]^B \right)$$

7.


This mixing correction uses surface measurements of the N₂O biological concentration ($[N_2O]^B$), the 'supply ratio' of oxygen saturation, given by the vertical gradient of biological O₂ to N₂O ($\frac{\partial [O_2]^B}{\partial [N_2O]^B}$), and the ratio of N₂O and O₂ gas transfer velocities ($\frac{k_{N_2O}}{k_{O_2}}$). Biological concentrations, indicated by the

superscript 'B', are derived by isolating and removing physical solubility effects from measured gas concentrations. The surface water biological concentration of N₂O, $[N_2O]^B$, was derived based on the difference between the N₂O saturation anomaly ($[N_2O]_{\text{sat}}$) and changes in N₂O solubility due to recent heat fluxes ($[N_2O]_{\text{meas}}$ - $[N_2O]_{\text{sat}}$ − $[N_2O]_{\text{thermal}}$). Heat flux effects on solubility, $[N_2O]_{\text{thermal}}$, were derived following Keeling and Shertz (1992), with corrections from Jin et al. (2017). Surface $[N_2O]_{\text{meas}}$ was

continuously measured from the surface seawater supply with an integrated cavity output spectroscopy (OA-ICOS) gas analyzer (Los Gatos Research, N₂O/CH₄ Analyzer, Model Number: 913–0055) coupled to a gas extraction module (Schuler and Tortell, 2023). The region-specific supply ratio, $\frac{\partial [O_2]^B}{\partial [N_2O]^B}$, was calculated by taking the slope of subsurface $[O_2]^B$ plotted against subsurface $[N_2O]^B$. The compiled data across the cruise track resulted in a supply ratio of $-1.6 * 10^4 \pm 0.3 * 10^4$ mmol O₂ (mmol N₂O)⁻¹,

which is similar to previous measurements for the Northeast Pacific ($-1.8 * 10^4$; Izett et al., 2018) and global basins ($-1.5 * 10^4$ ;(Cassar et al., 2014). Following Cassar et al. (2014), we assumed a constant $\frac{k_{N_2O}}{k_{O_2}}$ ratio of 0.92.

We note that several recent studies have observed nitrification within the euphotic zone, challenging the
assumption that N₂O production is limited to subsurface waters (Grundle et al., 2013; Smith et al., 2014), and potentially leading to overestimates in our vertical mixing-corrected NCP estimates. Previous observations in the CCS reported a range of depth-integrated mixed layer nitrification rates between $0.3 – 2$ mmol NH₄⁺ m⁻² d⁻¹, resulting in consumption of $0.6 – 4$ mmol O₂ m⁻² d⁻¹ (Stephens et al., 2020). Following the approach of Izett et al. (2018) we used a range of realistic N₂O:O₂

stoichiometries to estimate potential upper and lower bounds of mixed layer N₂O production. We determined mixed layer N₂O production likely ranged between $0.09 – 0.23$ $\mu$mol N₂O m⁻² d⁻¹, which would yield offsets in our final NCP estimates between 1.2 and 3.3 mmol O₂ m⁻² d⁻¹. Total uncertainty due to sources of error in other derived parameters was determined by following Izett (2021).

## 2.7 Nutrient concentrations

Samples collected during daily productivity casts were analysed for dissolved $NO_3^- + NO_2^-$, $PO_4^{3-}$, and silicic acid concentrations. Samples of 30 mL were collected from niskin bottles and filtered through GF/F filters, using acid-washed syringes into 20mL HDPE scintillation vials. Samples were kept frozen before analysis on a OI Analytical Flow Solutions IV auto analyzer by Wetland Biogeochemistry Analytical Services at Louisiana State University. Detection limits were 0.09 $\mu$mol $L^{-1}$ for nitrate + nitrate, 0.02 $\mu$mol $L^{-1}$ for phosphate and 0.02 $\mu$mol $L^{-1}$ for silicic acid. Reference standards for dissolved nutrients in seawater were used to ensure quality control.

Samples for iron analysis were collected with a rosette of Teflon-coated OTE-bottles during a separate cast directly after the daily sampling cast. After recovery, OTE-bottles were taken directly to a trace metal-clean sampling van where they were pressurized with filtered compressed air. Surface samples (~3 m depth) were also collected with a tow-fish system plumbed into the trace metal van, as in Bruland *et al.* (2005). All samples were passed through pre-cleaned 0.2 micrometer Supor membrane Acropak capsule filters into trace metal cleaned bottles (Cutter et al., 2014). Samples were acidified to pH 1.8 with optima HCl, and analyzed post-cruise with a flow injection analysis method (Lohan et al., 2006), with modifications as in Biller *et al.* (2013). Briefly, this method involved pre-concentrating the Fe at pH 2 with Toyopearl Chelate-650 resin and eluting into a reaction stream containing the colorimetric agent N,N-dimethyl-p-phenylenediaminedihydrochloride (DPD). The absorbance of the reaction stream was measured with a flow-through spectrophotometer. Calibration was performed with a standard addition curve, and blanks were assessed using acidified MilliQ. Reference samples analyzed to assess accuracy compared well to consensus values: SAFe D1 0.70 +/- 0.04 nmol Fe/kg, n=12 compared with consensus value 0.67 +/- 0.04 nmol/kg, and GEOTRACES GSC 1.51 +/- 0.07 nmol/kg (n=11) compared with consensus value 1.53 +/- 0.11 nmol/kg.

## 2.8 Transcriptomic analysis

Water collected from near-surface niskin bottle casts (46% surface irradiance) during daily productivity measurements was subsampled for RNA extraction. Approximately 2.5L to 4L of seawater were filtered onto 0.8 $\mu$m Pall Supor filters (142 mm) using a peristaltic pump, and then flash frozen in liquid nitrogen and stored at -80 °C. RNA was extracted using the RNAqueous-4PCR kit, following manufacturer instructions with the incorporation of a bead beating step during RNA lysis. All RNA samples were sent to GENEWIZ for library preparation and sequencing with PolyA tail selection. Sequencing was performed on an Illumina HiSeq 4000 with a 2x150 bp configuration. GENEWIZ provided raw paired-end read sequences for each sample.

Raw reads were trimmed using Trim Galore 0.6.10 (Martin, 2011) and quality control was determined with FastQC (Andrews, 2010). A *de novo* metatranscriptome assembly was conducted using rnaSPAdes 3.15.5 (Bushmanova et al., 2019) and CD-HIT-EST (Li and Godzik, 2006). Contigs were annotated using the Marine Functional Eukaryotic Reference Taxa (MarFERReT) database (Groussman et al., 2023), which provides NCBI taxonomic annotations (Federhen, 2012) and Pfam 34.0 functional

annotations (Mistry et al., 2021). Samples were mapped against the MarFERReT DIAMOND sequencing aligner and its compatible BLASTX command (e-value < 1e-06) (Buchfink et al., 2015). Trimmed samples were aligned using Salmon (Patro et al., 2017). The package tximport (Soneson et al., 2016) was used to generate a comprehensive table of read count data for each sample and each contig. Only counts taxonomically mapping to Bacillariophyta (i.e., diatoms) were included. The normalized counts for all genes were then calculated using DESeq2's median of ratios method (Love et al., 2014). Normalized counts of the low iron-inducible periplasmic protein (Fea1) (Allen et al., 2007), which shows high similarities to Iron Starvation-Induced Protein 2A (ISIP2A) (Behnke and LaRoche, 2020), was used as an indicator for iron stress (Marchetti et al., 2017).

## 2.7 Pigment Concentrations and Taxonomic Compositions

During daily productivity casts, duplicate 1L dark Nalgene bottles were filled with water from niskin bottles collected at 1% and 46% PAR level depths. Under low ambient light, samples were filtered onto 47mm GFF filters (Whatman, nominal pore size 0.7 $\mu$m). Filters were immediately flash frozen and stored in an onboard -80°C freezer. Samples were shipped on dry ice to the Pinckney Estuarine Ecology Photopigment Analysis Facility at the University of Southern California. There, photopigment concentrations were determined with high performance liquid chromatography following (Pinckney et al., 2001). Pigment compositions are reported in Appendix B.

In addition to pigment sampling, Light microscopy was used to identify and enumerate dominant phytoplankton taxa. For microscopic cell counts, 25-50 mL subsamples preserved in Lugol's solution were concentrated by sedimentation using Utermöhl chambers for >24h (Lund et al., 1958). Cell counts of recognizable dinoflagellate and diatom genera were carried out using an Olympus CKX-31 inverted microscope in at least ten fields of view per sample at 200x and 400x magnification.

## 3 Results

### 3.1 Oceanographic conditions

Across the cruise track, sea surface temperature (SST) ranged from 8.5 to 14 °C (Fig. 2), with strong coastal to offshore gradients (Fig. 1 and 2). The lowest SST was observed within near-shore upwelling plumes, which were associated with high salinity (>33). Along the entire cruise transect, salinity was negatively correlated with SST ($\rho$ = -0.73, $p \ll 0.01$), as expected for upwelling regions. Sharp hydrographic fronts were apparent along coastal to offshore transects. Moving offshore, SST rapidly increased, while salinity dropped, changing by as much as 2 degrees and 0.5, respectively, within a span of 5 km. These results indicate the presence localized on-shore upwelling plumes, as compared to more homogenous off-shore waters. Within the upwelling plumes, $NO_3^-$ concentrations were elevated, reaching maximum concentrations of 20.5 $\mu$M and displaying a positive relationship with salinity ($\rho$ = 0.89, $p \ll 0.01$) and a negative relationship with SST ($\rho$ = -0.76, $p \ll 0.01$). Off-shore, $NO_3^-$ decreased

to concentrations below the SUNA detection limit (~ 0.3 $\mu$M), highlighting the difference in nutrient availability between the oligotrophic offshore waters and productive coastal upwelling environments. Chlorophyll concentrations varied between 0.04 to 5.6 mg m$^{-3}$ and exhibited a statistically significant (though weak) positive relationship with NO$_3^-$ ($\rho = 0.30$, p << 0.01).

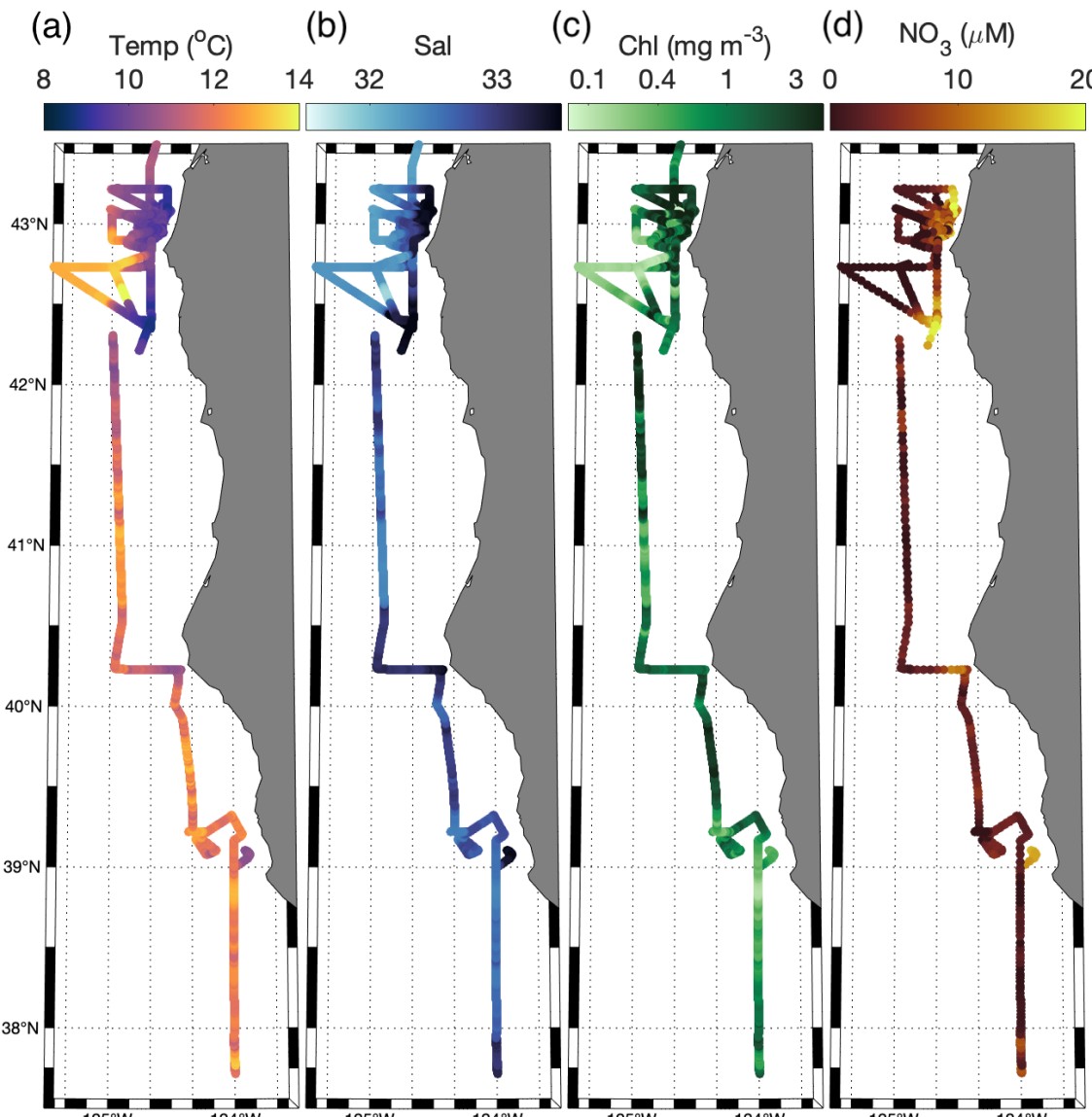

**Figure 2. Surface water (a) SST, (b) salinity, (c) Chl, and (d) NO$_3^-$ along the cruise track.** Cape Mendocino is abbreviated as Cape Mendo.

In addition to the coastal-offshore gradient, surface water hydrography also differed between the two distinct upwelling plumes we sampled. These plumes were identified as low SST in the lees of Cape

Blanco and Cape Mendocino (Fig 1). Both plumes exhibited an upwelling signature, but the apparent
intensity of upwelling (as reflected in SST, salinity, $NO_3^-$, and Chl) was significantly stronger within the
northern Cape Blanco plume (Fig. 3). Most apparently, SST was several degrees cooler at Cape Blanco
(median of $9.6 \pm 0.4$ °C) than at Cape Mendocino (median of $11.5 \pm 0.01$ °C). Nitrate concentrations
were highly variable across both plumes, but the mean $NO_3^-$ at Cape Blanco ($9.4 \pm 0.8$ $\mu$M) was nearly
twice as high as that of Cape Mendocino ($5.2 \pm 0.4$ $\mu$M). Chlorophyll concentrations were elevated at
both plumes relative to offshore waters, with a median of $1.9 \pm 0.01$ mg m$^{-3}$ around Cape Blanco and
$1.4 \pm 0.01$ mg m$^{-3}$ at Cape Mendocino. Both these Chl concentrations are well below that which can be
supported by the available $NO_3^-$ concentrations (1 $\mu$M $NO_3^-$ can typically yield 1 $\mu$g chl L$^{-1}$), indicating
that the phytoplankton blooms were likely in the early phases of development following upwelling.

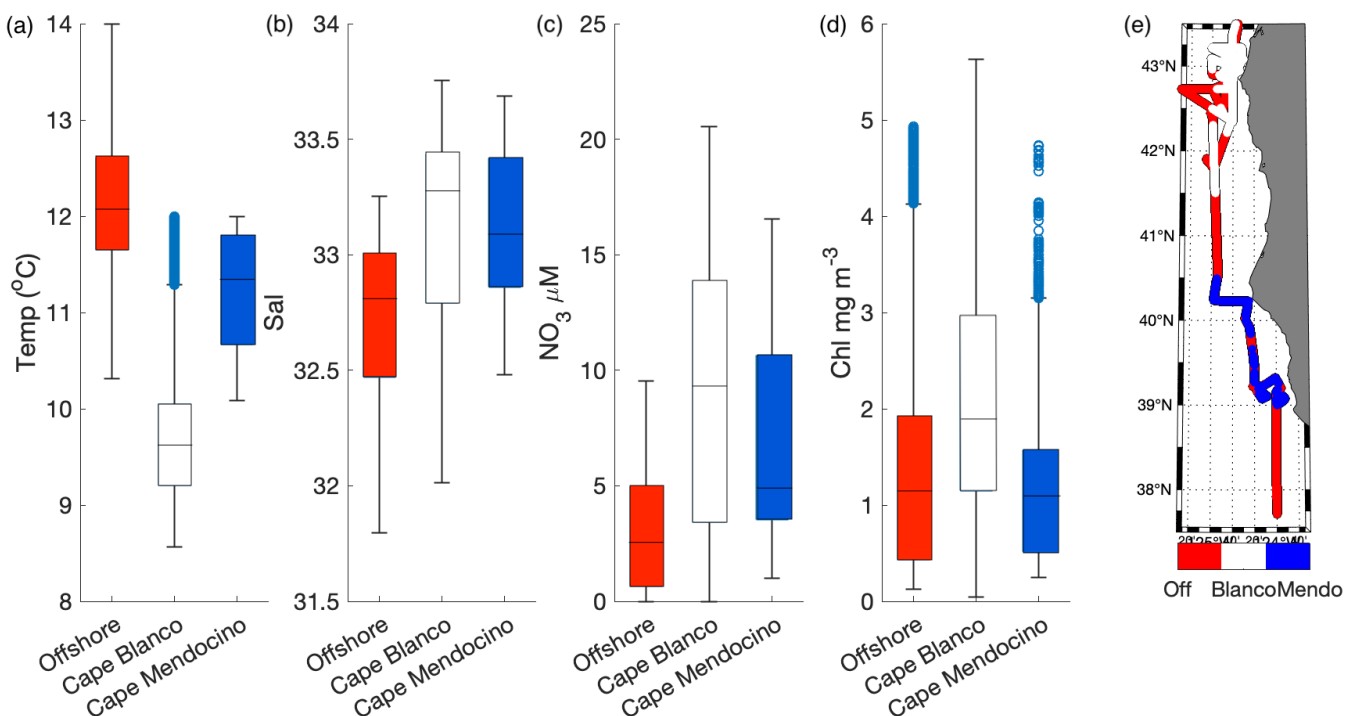


**Figure 3. Variability in SST (a), salinity (b), $NO_3^-$ (c), and Chl (d) within the observed water
masses offshore, at Cape Blanco, and Cape Mendocino**. The line inside each boxplot represents the
median, while whiskers display the 75$^{th}$ percentile. Points outside the whiskers represent outliers. Panel
(e) displays the spatial distribution of offshore (off), Cape Blanco, and Cape Mendocino (Mendo), data.
Samples with SST < 12°C and south of 41.5°N were considered part of the Cape Mendocino plume,
while samples with <12°C and north of 41.5°N were considered part of the Cape Blanco plume.

Samples falling out of these criteria were designated as offshore samples. Criteria were chosen based on a visual inspection of water hydrographic distributions (Fig 1.


Underway surface measurements were accompanied by on-station discrete sampling for $NO_3^-$, $PO_4$, Si, and Fe concentrations. As expected, $NO_3^-$ was highly correlated with $PO_4$ ($\rho = 0.99$, p $\ll$ 0.01) and $SiO_2$ ($\rho = 0.92$, p $\ll$ 0.01). The ratio of $NO_3^-$:$PO_4$ was 7.4 $\pm$ 2.0, and less than half the expected Redfield ratio of 16. This result is consistent with observations of low $NO_3^-$:$PO_4$ (~2-3) in the North

Pacific attributed to high subsurface denitrification rates (Tyrrell and Law, 1997). In contrast with the strong covariance observed among macronutrients, surface Fe distributions were not correlated with surface $NO_3^-$ concentrations ($\rho = 0.35$, p $= 0.6$). At Cape Blanco, Fe concentrations varied from 1.2 – 2.0 nM, whereas Cape Mendocino concentrations were significantly lower, ranging from 0.21 – 0.65 nM. Offshore Fe concentrations were relatively high, with a surface concentration of 0.29-0.42 nM.

These waters exhibited low macronutrient concentrations. In section 4.1 we explore the potential causes for different nutrient signatures in the various water masses we sampled.

**3.2 Photo-physiology**

Along the cruise track, phytoplankton photophysiological properties displayed spatial variability associated with hydrographic gradients, superimposed on significant diel cycles. In particular, we

observed strong diel signatures in the expression of various photo-protective mechanisms. Generally, we observed decreases in the PSII functional absorption area, $\sigma_{PSII}$, throughout the day, followed by recovery overnight (Fig. 4a). The maximum photochemical efficiency of PSII ($F_V/F_M$) similarly decreased during the day and peaked overnight. Measurements of $NPQ_{NSV}$ displayed an inverse diel pattern to those of $\sigma_{PSII}$ and $F_V/F_M$, reflecting adjustments to the allocation of absorbed energy between

competitive photochemistry and thermal dissipation pathways. However, the magnitude of diel variability in $F_V/F_M$, $\sigma_{PSII}$, and $NPQ_{NSV}$ signals displayed significant variability between subregions, as discussed below. These observations agree with previous diel cycle studies of the region (Schuback and Tortell, 2019).

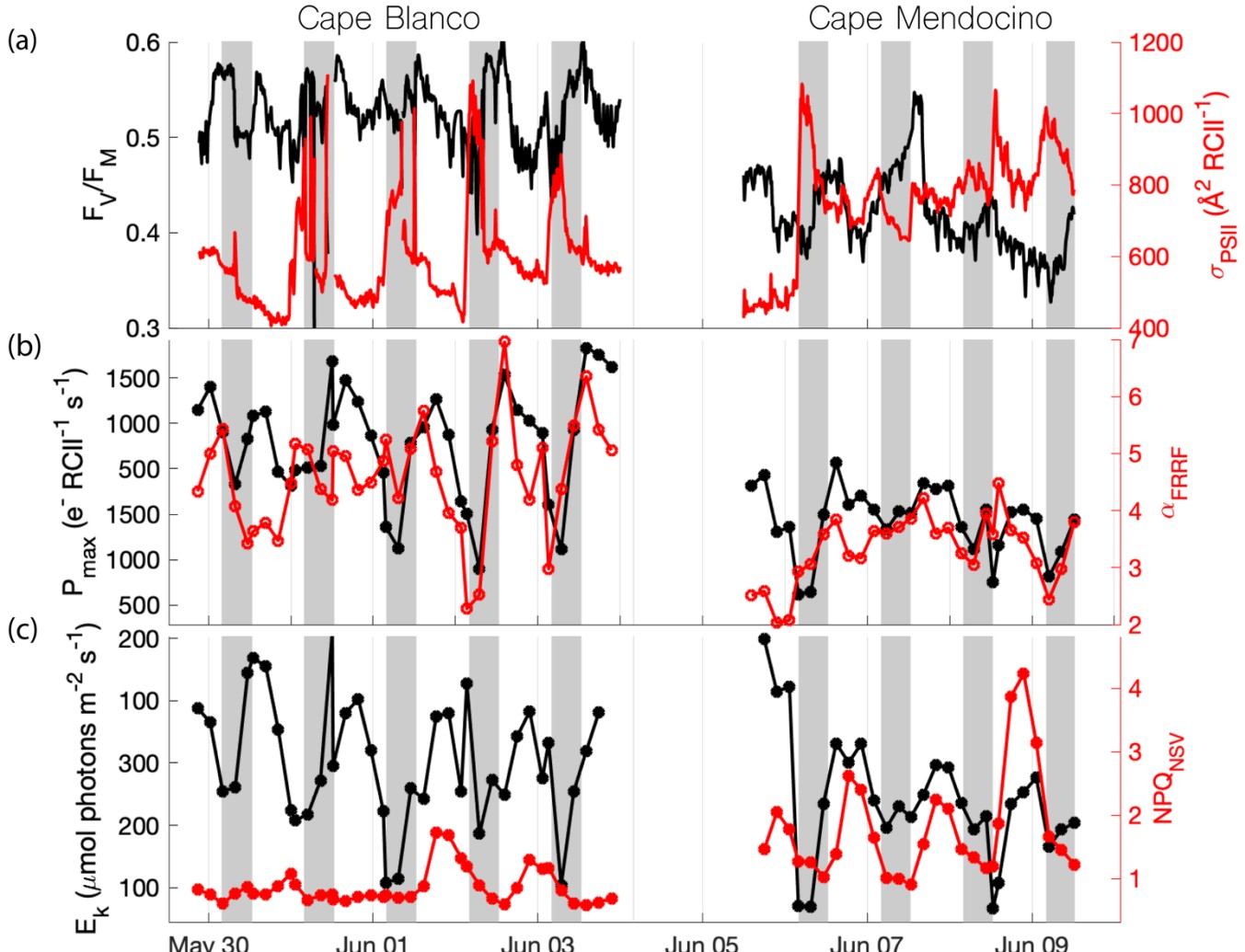

**Figure 4. Diel patterns in photo-physiological properties along the cruise track**. Grey shading indicates night-time, when surface PAR < 5 $\mu$mol photons m$^{-2}$ s$^{-1}$. **a.** $F_V/F_M$ (black) and $\sigma_{PSII}$ (red), **b.** $P_{max}$ (black) and $\alpha$ (red), **c.** $E_k$ (black) and NPQ$_{NSV}$ (red) are displayed with respect to sampling time. High sea states between June 4 - 6 impacted the seawater intake, resulting in the data gap.

Overall, photosynthetic parameters derived from semi-continuous PI curves also exhibited diel patterns that mirrored those of $F_V/F_M$ and $\sigma_{PSII}$. The maximum, light-saturated, ETR$_{PSII}$ ($P_{max}$) and the light utilization efficiency under light limiting conditions ($\alpha$) both peaked during the midday when *in-situ* irradiance was highest. The light saturation level, E$_k$, tracked surface light availability, while the photoinhibition parameter ($\beta$), data not shown, peaked during midday and decreased overnight (data not shown). As with the continuous underway data, results from these discrete PI curves match the previous diel observations of Schuback and Tortell (2019). We note, however, that there is potential for some convolution of temporal and spatial variability, as the ship spent more time offshore in the night, and

on-shore during the daytime. It is thus possible, that some of the diel cycling partially reflects variable photo-physiological signals between coastal and offshore waters.


Beyond diel signals, we also observed significant gradients in photophysiological parameters in relation to oceanographic conditions. In general, $F_V/F_M$, $P_{max}$, $E_k$ and $\alpha$ displayed positive relationships with upwelling indicators, i.e. high salinity and macronutrients, and low sea surface temperature (Table 1), suggesting that vertical transport of nutrient-rich water to the surface supported high photochemical

yields. In contrast, upwelling signals were associated with decreased $\sigma_{PSII}$ and $NPQ_{NSV}$. However, despite general trends between photo-physiological parameters and upwelling, there were significant differences in photo-physiological properties between the Cape Blanco and Cape Mendocino upwelling plumes. At Cape Blanco, mean values of $NPQ_{NSV}$ and $\sigma_{PSII}$ were significantly lower than at Cape Mendocino, while $F_V/F_M$, $P_{max}$, $\alpha$, and $\beta$ were all higher at this site, compared to Cape Mendecino

(Table 1). Photophysiological properties at Cape Mendocino were much closer to those observed in offshore non-upwelling waters, with mean $F_V/F_M$ values that were lower than offshore, despite elevated macro-nutrient concentrations. This result, combined with low Fe concentrations at Cape Mendocino, suggests that phytoplankton at Cape Mendocino were Fe-stressed despite the presence of upwelling conditions (see Discussion).


|  | CAPE BLANCO | CAPE MENDO | OFF-SHORE | SST N= 71604 | SAL N= 71604 | NO3- N= 989 | FE N=40 | SI N= 40 | PAR N= 71604 |
|---|---|---|---|---|---|---|---|---|---|
| $F_V/F_M$ N= 1438 | $0.47\pm0.01^A$ | $0.39 \pm 0.08^B$ | $0.41\pm 0.08^C$ | -0.55* | 0.27* | 0.39* | 0.32 | 0.52 | -0.04 |
| $\sigma_{PSII}$, N= 1438 | $545 \pm 5^A$ | $655 \pm 11^B$ | $647 \pm 5^C$ | 0.58* | -0.38* | -0.45* | -0.55 | -0.60 | -0.12 |
| $P_{MAX}$, N = 91 | $977\pm83^A$ | $600 \pm 42^B$ | $600 \pm 91^B$ | -0.67* | 0.40* | 0.50* | 0.45 | 0.78* | 0.19 |
| $\alpha$, N = 91 | $4.8\pm0.3^A$ | $3.3 \pm 0.3^B$ | $3.6 \pm 0.4^B$ | -0.57* | 0.19 | 0.24 | 0.37 | 0.45 | -0.11 |
| $\beta$, N = 91 | $1.55 \pm 0.47$ | $1.00 \pm 0.36$ | $0.98 \pm 0.43$ | 0.44* | 0.19 | 0.31 | 0.19 | 0.56 | 0.09 |
| $E_K$, N = 91 | $210\pm19^A$ | $172 \pm 20^B$ | $176 \pm 15^B$ | -0.38* | 0.33* | 0.41* | 0.62 | 0.72* | 0.37* |
| $NPQ_{NSV}$, N = 91 | $0.75\pm0.05^A$ | $1.33 \pm 0.2^B$ | $1.2 \pm 0.2^B$ | 0.66* | -0.26 | -0.46 | -0.36 | -0.62 | 0.58* |
| $F'_q/F'_m$ N = 91 | $0.21 \pm 0.10$ | $0.31 \pm 0.14$ | $0.33 \pm 0.13$ | 0.34* | -0.18 | -0.28 | -0.55 | -0.31 | 0.84* |

**Table 1. Summary of photophysiological properties at each sampling environment and their relationship to hydrographic parameters**. Left side of the table displays the median $\pm$ median absolute deviation for each photophysiological parameter according to sampling environment.

Superscripts denote groups with significantly different medians. Right side of the table displays

spearman rank correlation coefficients for each photophysiological parameter against environmental parameters. * indicates p < 0.05. Due to differences in sampling frequencies, the number of observations (N) varied across different parameters. For correlation analyses, paired observations were matched in space and time to the lowest resolution measurement. Cape Mendocino is abbreviated as Cape Mendo.

## 3.3 Primary Productivity

### 3.3.1 Electron Transport Rates, ETR$_{PSII}$

In-situ ETR$_{PSII}$, followed a notable diel cycle due to its first order dependency on irradiance (Eq 1, Fig. 5a). However, the relationship between PAR and ETR$_{PSII}$ was spatially variable across the cruise track, reflecting differences in $\alpha_{FRRF}$, P$_{max}$, and E$_k$ between Cape Blanco and Cape Mendocino (Table 1; Fig. 4). The amplitude of diel ETR$_{PSII}$ was greatest near Cape Blanco, despite lower average mixed layer PAR (Fig 5a). At Cape Blanco, maximum ETR averaged $977\pm83$ e$^-$ RCII$^{-1}$ s$^{-1}$, as compared to maximum values of only $600 \pm 42$ e$^-$ RCII$^{-1}$ s$^{-1}$ at Cape Mendocino. The higher maximum ETR at Cape Blanco is in good agreement with the observations of higher photochemical yields in this region (Table 1).

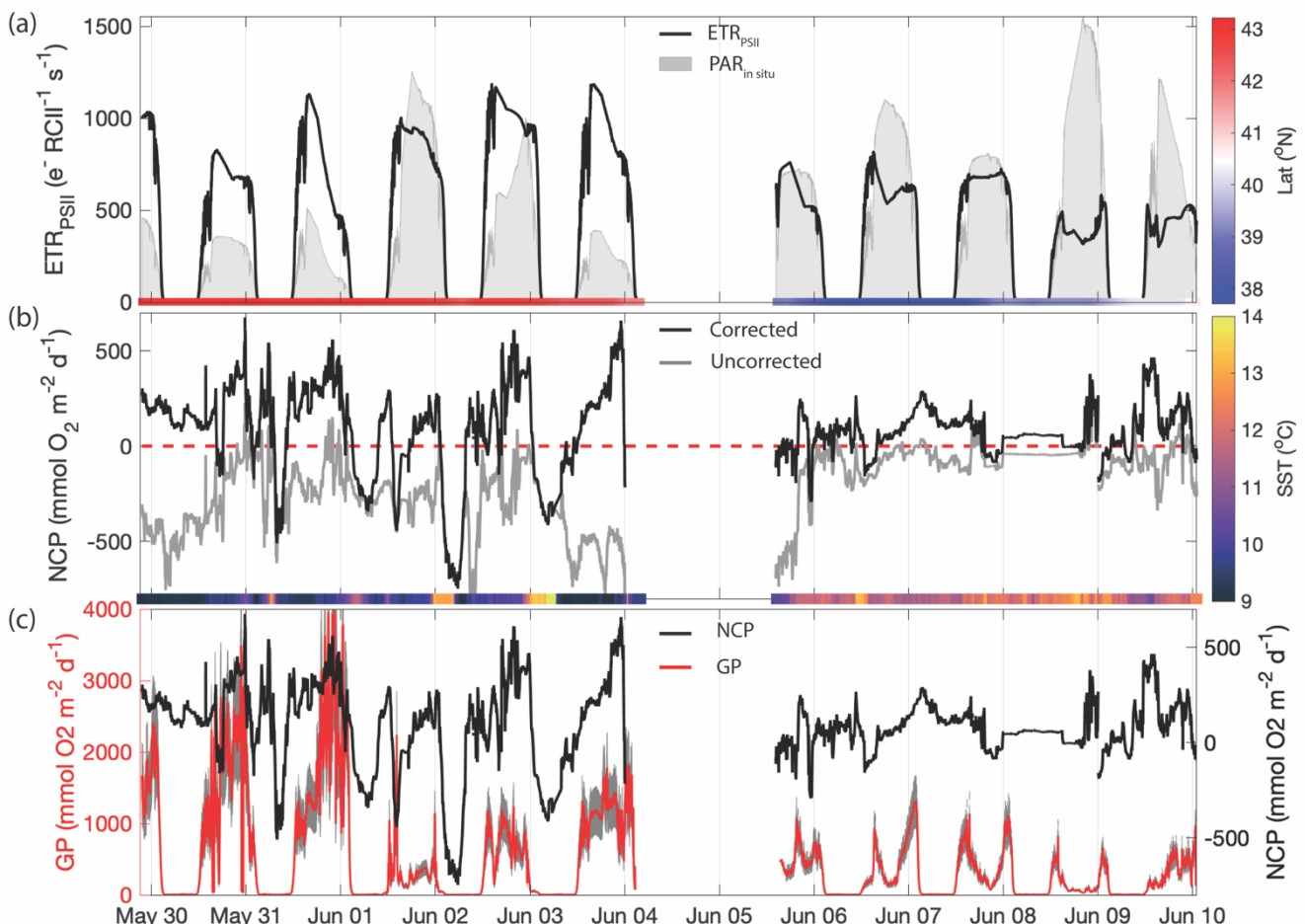

**Figure 5. Primary productivity time series along the cruise track. a)** ETR$_{PSII}$ derived from FRRF-measured PI curve parameter interpolated to match NCP sampling frequency (black line). The grey patches indicate the mean PAR available within the mixed layer (PAR$_{in\ situ}$). The bar on the bottom is coloured by the sampling latitude. The Cape Blanco filament is designated as latitude > 40.5 and is coloured red, while Cape Mendocino is designated as latitude < 40.5 coloured blue. **b)** Uncorrected NCP (grey) and mixing-corrected NCP (black). The dashed red line denotes the boundary between net autotrophy (NCP > 0) and net heterotrophy (NCP < 0). The color bar on the bottom illustrates the sample temperature. **c)** mixing-corrected NCP (black, right y-axis) and gross photochemistry (GP), shown in red (left y-axis). GP is calculated by converting ETR$_{PSII}$ from units e$^-$ RCII$^{-1}$ s$^{-1}$ to mmol O$_2$ m$^{-2}$ d$^{-1}$. The grey shading around the red line displays the range of GP based on an assumed range of 400 – 700 Chl:RCII.

### 3.3.2 Net Community Productivity, NCP

In addition to GPP, we estimated NCP from underway measurements. Prior to correcting for vertical mixing, more than 80% of derived NCP values were negative, suggesting net heterotrophic conditions over most of the cruise track. The most negative uncorrected NCP values were observed near Cape Blanco, despite the high gross photochemistry rates measured in this region (Fig 5a and b).

The apparent decoupling between NCP and GPP can be largely explained by vertical mixing of low $O_2$ waters, which artificially depress $O_2$-derived NCP estimates (see Sect. 2.4). After applying the $N_2O$-based mixing correction, we found that the majority of the cruise track (73% of measurements) exhibited net autotrophy, with the highest value recorded within the Cape Blanco filament. For the most part, net heterotrophy only existed at night in warmer off-shelf waters. The mean corrected NCP was 80

$\pm$ 218 mmol $O_2$ $m^{-2}$ $d^{-1}$, within range of previous observations of late-spring NCP within the California Current (Kranz et al., 2020). The large standard deviation reflects the large diel and spatial variability observed along the cruise track. The highest NCP estimates we obtained (> 500 mmol $O_2$ $m^{-2}$ $d^{-1}$) are on the upper end of previous measurements. Values above 100 mmol $O_2$ $m^{-2}$ $d^{-1}$ have only been observed in the most productive coastal waters (Wang et al., 2020; Niebergall et al., 2023), further emphasizing

the high productivity of the CCS.

The $N_2O$-derived mixing correction term was strongly correlated to $N_2O$-independent indicators of upwelling, namely temperature ($\rho$ = -0.77, p << 0.01). Uncertainty in vertical-mixing corrected NCP due potential mixed layer nitrification (see sect. 2.6) represented between 1.5 – 4.2% of our mean

corrected NCP value. These results give confidence that high surface concentrations of $N_2O$ are a valid marker of upwelling and transport of $O_2$-poor subsurface water into the mixed layer. The maximum correction factor, 1200 mmol $O_2$ $m^{-2}$ $d^{-1}$, was observed within the cold upwelling filament near Cape Blanco, where uncorrected NCP was below -500 mmol $O_2$ $m^{-2}$ $d^{-1}$. This result highlights the impact of vertical fluxes on $O_2$-based NCP estimates in upwelling regions.

### 3.3.3 Carbon fixation rates

At nine discrete sampling stations, $^{14}C$-based PI curves were measured in parallel with $ETR_{PSII}$ at the surface and at the base of the euphotic zone. Volumetric carbon fixation rates varied significantly between stations and depths. Maximum carbon fixation rates (carbon-based $P_{max}$) ranged from 0.4 to 96 $\mu g$ C $L^{-1}$ $hr^{-1}$. Over 85% of the variability in carbon fixation rates could be explained by differences in

biomass, which varied from 0.11 to 9 mg Chl $L^{-1}$. Chlorophyll concentrations in near-surface waters were, on average, four times higher than those at the base of the euphotic zone, implying the bulk of carbon fixation took place in the mixed layer. On average, the bottom depth of the euphotic zone was 14 $\pm$ 12 meters deeper than the bottom depth of the mixed layer

To compare carbon-based GPP estimates against parallel $ETR_{PSII}$ measurements, carbon fixation rates were normalized to chlorophyll and converted to units of C $chl^{-1}$ $s^{-1}$. Chlorophyll normalized carbon

fixation rates were positively correlated with Fe ($\rho = 0.40$, p < .05), $F_V/F_M$ ($\rho = 0.56$, p < .05), and Si:$NO_3^-$ ratios ($\rho = 0.60$, p < .05), and negatively correlated with $\sigma_{PSII}$ ($\rho = -0.68$, p < .05). These results suggest that stations with low chlorophyll-normalized carbon fixation rates may have been affected by

Fe and Si co-limitation. By comparison, carbon fixation rates were not significantly correlated with $NO_3^-$ or $PO_4$ concentrations or salinity.

Notably, carbon fixation consistently saturated at lower light intensities than $ETR_{PSII}$. The average $E_k$ for $ETR_{PSII}$ was $5 \pm 2.8$ times greater than $E_k$ for carbon fixation. As a result, carbon fixation did not

scale linearly with $ETR_{PSII}$, but rather demonstrated a hyperbolic relationship at each station and depth sampled. At sub-saturating light levels, carbon fixation increased linearly with $ETR_{PSII}$, until approaching an asymptote as $ETR_{PSII}$ continued to increase while carbon-fixation remained stationary (Fig 6). This result indicates that at light levels beyond the saturating index for carbon-fixation, $ETR_{PSII}$ provides reducing power in excess Calvin-Benson Cycle requirements. Previous studies have noted a

similar nonlinear relationship between carbon fixation and $ETR_{PSII}$, consistent with an upregulation of alternative electron pathways under high light levels (Schuback and Tortell, 2019; Suorsa, 2015; Zhu et al., 2017). This non-linear relationship between C fixation and ETR has been cited as a key limitation to the widespread use of FRRF for autonomous high resolution GPP estimates. Although our results demonstrate a clear hyperbolic relationship between these rates, the parameters describing this

relationship were variable across stations. In sect. 4.3, we further examine the relationship between carbon fixation and $ETR_{PSII}$, and contextualize the apparent differences in carbon-electron decoupling with available physiological, and environmental data.

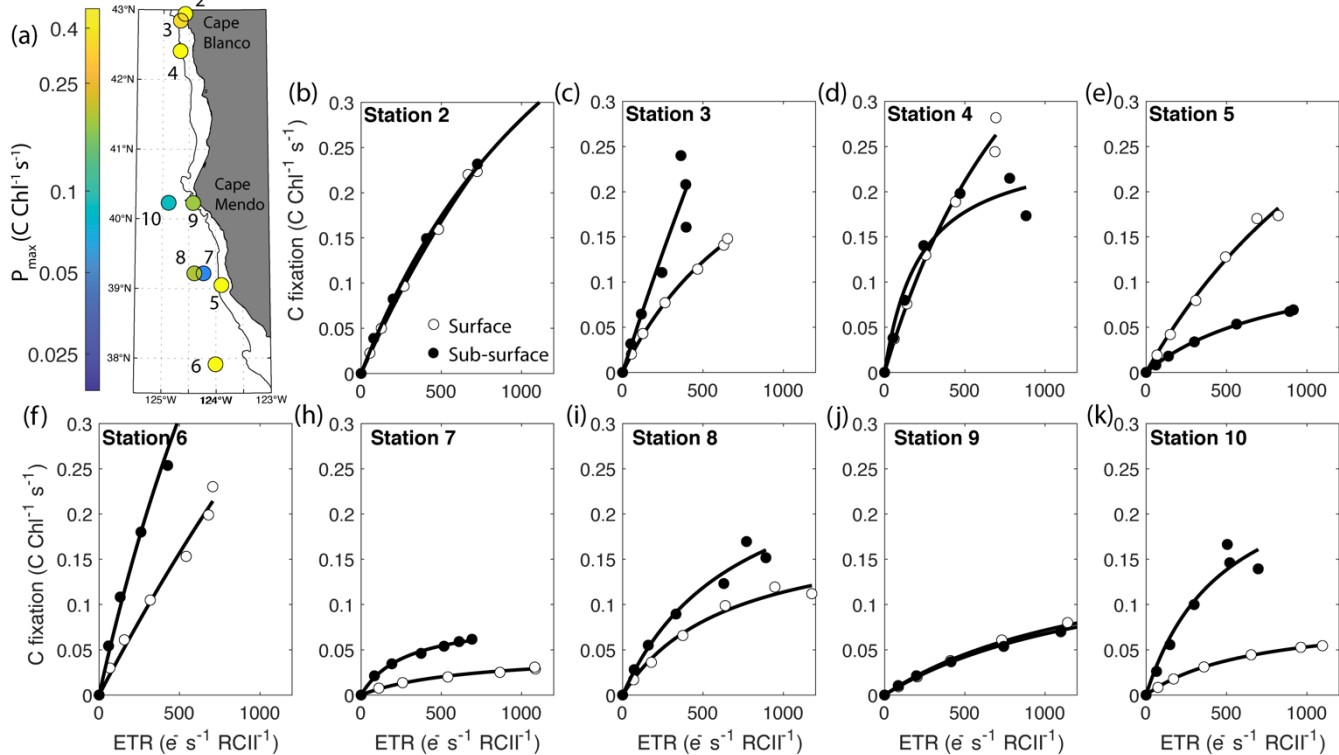


**Figure 6. Parallel measurements of fluorescence- and ¹⁴C-based photosynthesis-irradiance experiments conducted on-station.** The top left displays the sample locations coloured by maximum chl-specific surface carbon fixation rates. The black contour line displays the 200m isobath. The other panels show carbon fixation rates plotted against $ETR_{PSII}$. Surface samples are represented by open

circles while sub-surface samples are depicted as closed circles. Each curve is fit with the function,

$$C\ fixation = \frac{P_{max-C} * ETR_{PSII}}{K_{sat} + ETR_{PSII}},$$ where $P_{max-C}$ is the maximum carbon fixation rate, and $K_{sat}$ is the

saturation constant beyond which changes in carbon fixation with respect to $ETR_{PSII}$ become increasingly non-linear. Carbon fixation data were not collected at Station 1. All curve fits had an $R^2 > 0.9$.

**4 Discussion**

As expected, our continuous underway measurements revealed strong spatial and temporal variability in biogeochemical properties across the California Upwelling system. In particular, we observed large diel cycles, and coastal-offshore gradients in biogeochemical properties, with two distinct upwelling filaments in the vicinity Cape Mendocino and Cape Blanco. Differences in nutrient availability between

sample sites appeared to exert a strong influence on photo-physiology, gross photochemistry, gross primary productivity and net community productivity. In this section, we explore the potential underlying causes of biogeochemical differences across our survey region, with a focus on iron gradients across the two distinct upwelling filaments. We also discuss the direct and indirect influence

of iron availability and other environmental variables on phytoplankton photo-physiology, and energy
transfer efficiencies between photosynthetic processes.

**4.1 Factors driving the contrasting biogeochemistry of Cape Blanco and Cape Mendocino filaments**

The significant differences in surface oceanographic conditions between upwelling filaments may have been driven by differences in 1) the strength and timing of upwelling at the two Capes, or 2) differences in the nutrient content of the sub-surface upwelling source waters. We investigated these two
possibilities by examining the NOAA coastal upwelling transport index (CUTI) as a proxy for upwelling strength during and prior to the sampling period, and by evaluating nutrient depth profiles to examine the upwelling source waters at the two capes. Our analysis suggests that both factors likely contributed to the apparent differences between Cape Blanco and Cape Mendocino biogeochemistry, providing evidence that Fe and Si concentrations were particularly affected by bathymetric features that
influence Fe supply.

**4.1.1 Strength of upwelling and relative age of filaments**

To examine differences in the timing and strength of upwelling between the two filaments, we tracked CUTI for a 10 day interval prior to sampling (Jacox et al., 2018). Throughout the sampling period, the strength of upwelling at Cape Blanco varied from 1.3 to 2.4 m $d^{-1}$ (positive values indicate upwelling)
and peaked 6 days prior to our arrival, at 3.8 m $d^{-1}$. During our sampling period at Cape Mendocino, upwelling varied from -0.2 to 3.3 m $d^{-1}$. Upwelling conditions persisted in this region, with vertical transport rates > 2 m $d^{-1}$ between June 1 and 4, up to one day prior to our arrival. However, after June 4, the vertical mixing index at Cape Mendocino rapidly reversed to weak downwelling (-0.1 to -0.3 m $d^{-1}$) during the last few days of sampling, emphasizing the dynamic nature of this sampling environment.
These results support the hypothesis that colder and more nutrient rich water near Cape Blanco was attributable to stronger and more consistent upwelling in this region in the interval prior to our sampling. In contrast, Cape Mendocino was transitioning from upwelling to downwelling conditions during our sampling period.

**4.1.2 Nutrient Content of upwelling source waters**

Depth profiles offer additional insight into the nutrient concentrations of upwelling filament source waters. Unfortunately, nutrient samples were only collected down to the base of the euphotic zone, missing the deeper source waters. Nonetheless, measurements at the base of the euphotic zone (40-50m) enable us to compare subsurface nutrient concentrations. Mean concentrations of $[NO_2^- + NO_3^-]$ between 40 and 50m were significantly ($p < 0.05$) higher at Cape Blanco ($24.5 \pm 4.3\ \mu M$) than at Cape
Mendocino ($16.5 \pm 5.7\ \mu M$). Similarly, phosphate concentrations between 40-50m were significantly greater around Cape Blanco ($2.6 \pm 0.3\ \mu M$) compared to Cape Mendocino ($2.1 \pm 0.4\ \mu M$; $p = 0.02$). Relative to nitrate and phosphate, even larger differences were observed in Fe and Si concentrations

between the two capes. The 40-50m silicic acid concentration at Cape Blanco ($30.0 \pm 8.6 \, \mu M$) was nearly double that observed at Cape Mendocino ($16.3 \pm 8.2 \quad \mu M$; ($p \ll 0.01$), while, Fe concentrations between 40-50m at Cape Blanco ($6.8 \pm 4.1$ nM) were more than three-fold higher than those at Cape Mendocino ($1.8 \pm 2.3$ nM; $p = 0.01$). These results support the hypothesis that the two upwelling plumes were seeded by different water masses with distinct nutrient concentrations.

Differences in underlying bathymetric features between Cape Blanco and Mendocino likely contributed to the observed differences in Fe and Si availability. Cape Blanco sits over a broad section of the continental shelf (> 30km wide) composed of highly erodible sedimentary rocks with mineral rich sand-silt layers originating from the Klamath Mountains (Spigai, 1971). The broad shelf continues south until the triple junction of the North American, Pacific, and Gorda plates which forms the submarine Mendocino escarpment, a narrow ridge extending west from Cape Mendocino along the transform fault (Menard and Dietz, 1952). Importantly, the shelf rapidly narrows to less than 5 km at the latitude of Cape Mendocino (Appendix A). Differences in shelf width have important implications on sub-regional iron availability. Previous work by Biller *et al.* (2013) demonstrated that shelf width correlated with greater Fe bioavailability in the water layer directly overlying the seafloor. This trend was evident in our study as well, with a positive correlation ($\rho = 0.57$) between shelf width and Fe concentrations at the bottom of the euphotic zone for on-shelf stations. Yet, with only nine stations on-shelf stations, this correlation was not statistically significant.

In contrast with $NO_3^-$ and $PO_4^{3-}$, which are resupplied to the surface by upwelling, remineralized Fe is rapidly removed from the water column, such that Fe supply to the surface can be significantly decoupled from macronutrients (King and Barbeau, 2011; Bruland et al., 2014). As a result, differences in upwelling strength at Cape Blanco and Cape Mendocino likely account for the differences in $PO_4^{3-}$ and $NO_3^-$ between the two sites, while contrasting shelf features can explain the larger differences in Fe availability. These observations fit within the theory that the California Current contains a 'mosaic' of Fe limitation, where patches of Fe-poor water may persist even in the presence of upwelling conditions (Till et al., 2019; Hutchins et al., 1998). In the following section, we present several lines of evidence that photo-physiological properties of phytoplankton assemblages were influenced by iron gradients.

## 4.2 Environmental and taxonomic influences on physiology and productivity

Environmental gradients exert direct effects on phytoplankton physiology and productivity by determining the supply of essential nutrients that support cellular growth and the maintenance of photosynthetic proteins. As a cofactor in many biological redox reactions, Fe plays a particularly important role in the photosynthetic electron transport chain and nutrient uptake pathways. Several lines of evidence suggest that iron stress was a key factor shaping phytoplankton productivity and photo-physiology across our study site. As noted above, there was a significant difference in Fe concentrations between Cape Blanco (high Fe) and Mendocino (low Fe) associated with variability in the shelf width. The difference in Fe-availability was strongly correlated with $Si:NO_3^-$ ($\rho = 0.85$, $p \ll 0.0.1$), likely reflecting excess Si uptake by iron-limited diatoms (Sarthou et al., 2005; M. Franck et al., 2000).

Further evidence of Fe-stress at Cape Mendocino was obtained from ancillary transcriptomic analysis, which demonstrated elevated expression of the Fe assimilation gene *Fea1* (Appendix C), which has previously been cited as a marker of Fe-stress (Allen et al., 2007). Together, these observations indicate the onset of Fe stress at sites with reduced Fe-availability.


Beyond direct effects, Fe gradients indirectly influence phytoplankton physiology and productivity by driving taxonomic shifts towards species that are adapted to low Fe conditions. Although diatoms were the most dominant group across the study area, their relative contribution was significantly lower around Cape Mendocino compared to Cape Blanco (Appendix B). At Cape Mendocino, we observed a taxonomic shift towards smaller phytoplankton, including smaller diatoms and dinoflagellates. Smaller cells sizes afford larger surface area to volume ratios, facilitating nutrient uptake at lower concentrations (Sunda and Huntsman, 1997). Moreover, stations near Cape Mendocino had high abundances of *Pseudo-nitzschia*, whereas this genus was not observed near Cape Blanco. *Pseudo-nitzschia* is a well-studied diatom with a number of physiological adaptations to Fe-limitation (Lampe et al., 2018). These shifts in phytoplankton assemblages towards smaller sizes and low Fe specialists suggest bottom-up environmental controls driving taxonomic composition.



Cell size and nutrient status influence the optical properties and photo-physiology of phytoplankton. Large cells are prone to pigment packaging effects, which decrease Chl-specific absorption as intracellular Chl concentrations increase and surface area to volume ratios decrease. This effect causes reduced $\sigma_{PSII}$, as was observed near Cape Blanco (Table 1). Nutrient limitation, particularly for Fe, can also lead to accumulation of photo-inactive or damaged RCII, which still absorb light but do not contribute to photochemistry (Roncel et al., 2016). This further drives high $\sigma_{PSII}$, which is proportional to the light harvesting complex absorption coefficient normalized by active RCII concentrations (Oxborough et al., 2012; Z. Li et al., 2021), and low $F_V/F_M$, due to inactive RCII contributing to the $F_M$ but not $F_V$ signal (Schuback et al., 2021). Both of these commonly cited indicators of Fe stress were observed around Cape Mendocino and in some offshore regions (Table 1).



The taxonomic and nutrient-dependent effects on photo-physiology described above directly impact $ETR_{PSII}$ (Eq. 1). Previous studies have noted higher $ETR_{PSII}$ among Fe-limited phytoplankton, presumably due to increased $\sigma_{PSII}$ (Schuback *et al.*, 2015). However, we observed greater $ETR_{PSII}$ in the relatively Fe-rich waters near Cape Blanco (Table 1), likely due to high $F_q'/F_v'(PAR)$, which represents the proportion of open RCII at a given light level (Suggett et al., 2011). Low NPQ observed in the Cape Blanco region (Fig. 4d) likely enabled $F_q'/F_v'$ to remain high under high light levels. Indeed, $F_q'/F_v'$ measured during underway Photosynthesis-Irradiance curves and interpolated to in-situ irradiances demonstrated that $F_q'/F_v'$ was higher around Cape Blanco compared to Cape Mendocino (Table 1) It is well recognized that iron limitation exacerbates high light stress and NPQ (Schallenberg et al., 2020; Ryan-keogh et al., 2020), and the high NPQ at Cape Mendocino compared to Cape Blanco provides further evidence that Cape Mendocino assemblages were affected by Fe stress. Iron limitation also impacts photosynthetic processes downstream of $ETR_{PSII}$. In this study, maximum carbon fixation rates




($P_{max}$ determined during $^{14}C$ PI experiments) displayed a strong correlation with Si ($\rho = 0.63$, p $<<$ 0.01) and the ratio of $Si:NO_3^-$ ($\rho = 0.60$, p $<<$ 0.0.1) in the water column.

Overall, our results suggest that Fe availability gradients between Cape Blanco and Mendocino influenced local community composition and physiology with consequential effects on C and Si cycles. Differences in taxonomic composition, photo-physiology, nutrient quotas and productivity all serve as evidence that the community shifted towards Fe-limitation in proximity to Cape Mendocino. Due to the differential sensitivity of $ETR_{PSII}$, carbon fixation, and NCP to Fe-availability, we hypothesized that Fe-

limitation would lead to a decoupling between these different PP currencies. We explore this hypothesis below with direct comparisons of $ETR_{PSII}$, C-fixation, and NCP.

### 4.3 Energy transfer efficiencies between photosynthetic processes

Measurements of primary productivity in different 'currencies' (electrons, carbon, and oxygen) reflect the rates of distinct photosynthetic processes. Comparison of these alternative productivity metrics can

yield information on energy transfer efficiencies across the photosynthetic pipeline (Fig 7). A minimum of four charge separation events are required to produce one $O_2$ and fix one $CO_2$, stoichiometrically linking water splitting and carbon fixation. Yet a number of non-linear electron transport pathways can divert reducing power from carbon fixation, decoupling $ETR_{PSII}$ from GPP, while oxygen consumption by respiration and non-linear electron transport pathways can further decouple $ETR_{PSII}$ from NCP (Fig

7). The ratio between $ETR_{PSII}$ and GPP thus provides information on the magnitude of non-linear electron transport, while the ratio between $ETR_{PSII}$ and NCP reflects the sum of non-linear electron transport and respiration. In practice, interpreting the decoupling between ETR, GPP, and NCP is complicated by differences in the temporal and spatial scales of the various measurement approaches, as well as the different assumptions implicit in each method. In the following sections, we directly

compare parallel productivity measurements to examine energy transfer efficiencies across photosynthetic processes, taking care to note important methodological considerations.

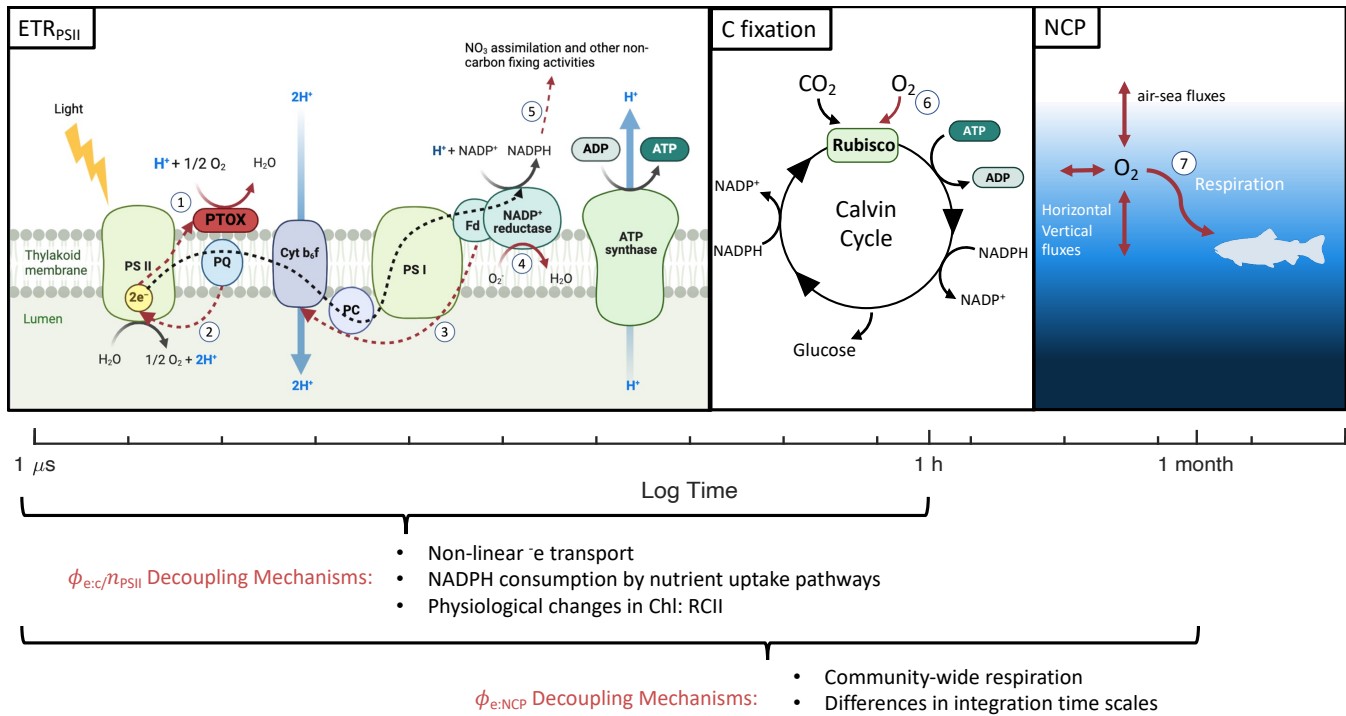

**Figure 7. Different primary productivity currencies and their decoupling mechanisms.** The three
productivity currencies of interest are indicated with respect to the times scales they represent.
Numbered red pathways denote decoupling mechanisms including 1) chloro-respiration 2) cyclic
transport around PSII 3) cyclic transport around PSI 4) pseudo-cyclic/ Mehler reactions 5) reductant
consuming nutrient uptake pathways 6) photo-respiration and 7) Community-wide respiration. Figure
produced in Biorender.

### 4.3.1 Carbon fixation as a function of ETR_PSII

Non-linear electron transport pathways (Fig 7) can act to maintain redox homeostasis when $ETR_{PSII}$
exceeds downstream energy requirements for growth and metabolism. The energy balance between PSII
and PSI becomes disrupted under high irradiance, when PSII absorbs energy in excess of PSI electron
transport rates, and/or under nutrient limitation, when biosynthesis of electron transporters is limited
(Hughes et al., 2018b; Schuback et al., 2015; Roncel et al., 2016). Iron limitation, in particular, exerts
acute constraints on the synthesis of Photosystem I (PSI) and Cytochrome $b_6f$ (Cyt $b_6f$), which require
12 and 5 Fe atoms each (Raven et al., 1999). As a result, Fe-limited phytoplankton have high levels of
PSII relative to PSI, exacerbating energy imbalances between PII and PSI, and necessitating
760 upregulation of non-linear electron transport pathways (Behrenfeld and Milligan, 2013). We therefore
hypothesized that Fe-stress would increase decoupling between C-fixation and $ETR_{PSII}$.

One of our primary findings is that C-fixation varies as a hyperbolic function of $ETR_{PSII}$, with curves defined by the maximum carbon fixation rate ($P_{max-C}$) and the saturation constant ($K_{sat}$). These parameters were highly variable between samples (Fig. 6). Samples with high $P_{max-C}$ and $K_{sat}$, (e.g. Station 6) showed more linear relationships between C-fixation and $ETR_{PSII}$, and a nearly constant electron requirement for carbon fixation ($\phi_e : c/n_{PSII}$, units = $^-$e Chl C$^{-1}$ RCII$^{-1}$). In contrast, in phytoplankton assemblages with low $P_{max-C}$ and $K_{sat}$, carbon-fixation quickly saturated with respect to $ETR_{PSII}$, resulting in an increase in $\phi_e : c/n_{PSII}$ with increasing light levels (e.g. Station 7). Determining sources of $P_{max-C}$ and $K_{sat}$ variability therefore provides significant utility in predicting electron requirements for gross carbon fixation, a key parameter required for fluorescence-based GPP measurements.

Previous studies have documented the importance of different environmental, taxonomic and physiological parameters in driving variability in $\phi_e : c/n_{PSII}$, but efforts to develop empirical algorithms predicting $\phi_e : c/n_{PSII}$ remain ongoing (Lawrenz et al., 2013). Recent progress has been made on this front by Schuback *et al.*, (2015, 2016, 2017) who reported a consistent relationship between $NPQ_{NSV}$ and $\phi_e : c/n_{PSII}$ in the surface waters of the Northeast Pacific and Canadian Arctic, suggesting that carbon fixation can be estimated using FRRF-based NPQ and $ETR_{PSII}$ measurements alone. Subsequent studies have applied the $NPQ_{NSV} \propto \phi_{e:C}/n_{PSII}$ relationship observed by Schuback et al. (2015) to collect high-resolution fluorescence-based GPP estimates, e.g. Kranz *et al.* (2020). Others, however, have noted that the $NPQ_{NSV} \propto \phi_{e:C}/n_{PSII}$ relationship does not hold for all taxa (Hughes et al., 2021), light conditions (Schuback et al., 2017), or environments where phytoplankton grow on more reduced N forms like $NH_4^+$ (Fei et al., 2024).

In our study, $NPQ_{NSV}$ was positively correlated with $\phi_{e:C}/n_{PSII}$ ($\rho = 0.55, p \ll 0.01$). However, the linear relationship proposed by Schuback et al. (2017; 2019) did not adequately predict $\phi_{e:C}/n_{PSII}$ for our samples ($R^2 = -0.41$), nor did we find a single line of best fit that could describe all of our data ($R^2 = 0.30$). Rather, we found that $\phi_{e:C}/n_{PSII}$ scaled directly with $NPQ_{NSV}$ only for samples with low $P_{max-C}$ (=< 0.3 C Chl$^{-1}$ s$^{-1}$). In contrast, for samples with high $P_{max-C}$ (> 0.3 C Chl$^{-1}$ s$^{-1}$), $\phi_{e:C}/n_{PSII}$ remained relatively constant across increasing light, $NPQ_{NSV}$, and $ETR_{PSII}$ (Fig 8).

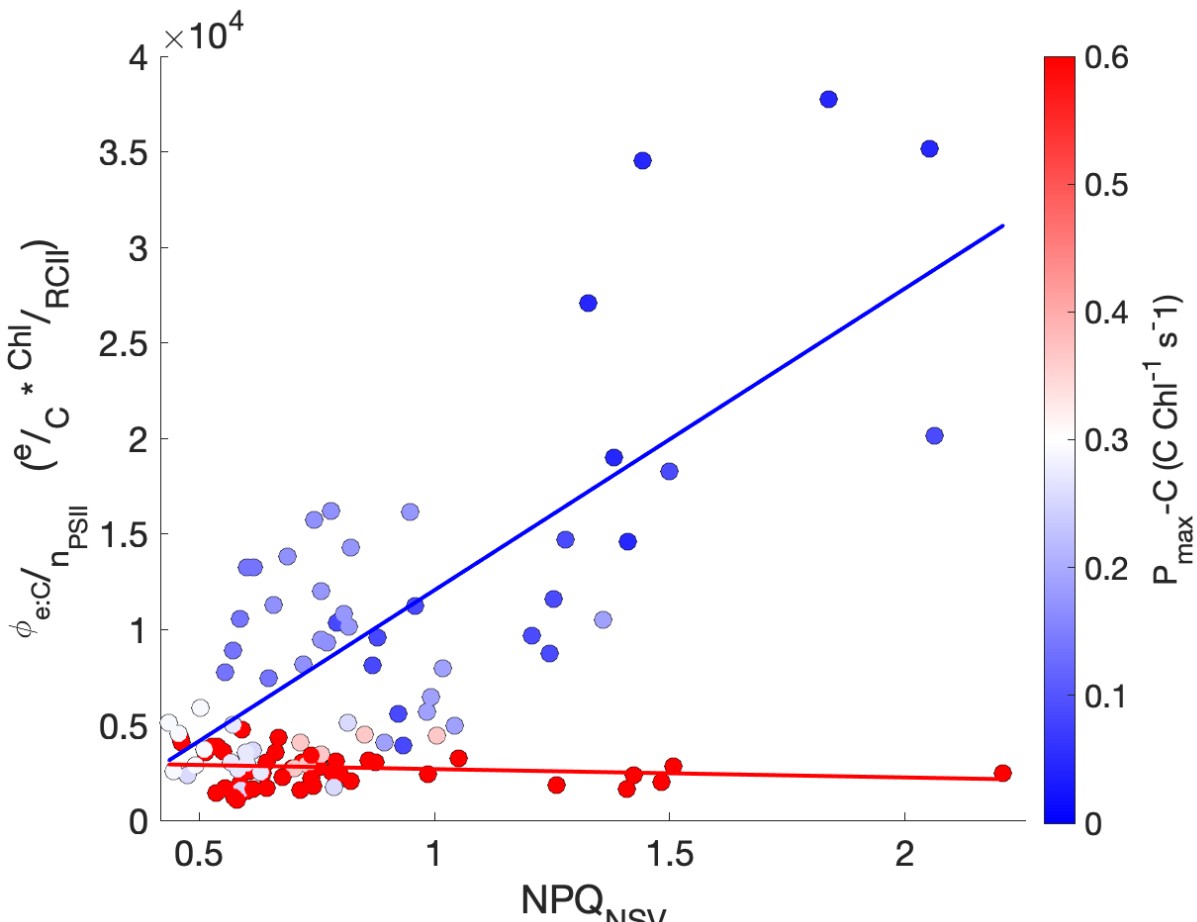

**Figure 8. The relationship between $\phi_{e:C}/n_{PSII}$, NPQ$_{NSV}$ and P$_{max-C}$.** The electron requirement for carbon fixation, $\phi_{e:C}/n_{PSII}$, measured as the ratio of ETR$_{PSII}$ to Chl-normalized carbon fixation, is plotted against NPQ$_{NSV}$. Each data point is colored by the sample's P$_{max-C}$, shown by the red/blue colorbar. Lines of best fit are drawn through points with P$_{max}$-C <= 0.3 and > 0.3 mol C mol Chl$^{-1}$ s$^{-1}$, with the colorbar indicating P$_{max}$-C.

By analogy with the Michaelis-Menten enzyme kinetics model, variability in P$_{max-C}$ can be explained as the product of the enzyme concentration and maximum reaction rate (Choi et al., 2017). In our case, P$_{max-C}$ reflects the entire suite of proteins that facilitate the conversion of chemical energy to organic matter. Reduced concentrations of PSI and Cyt b$_6$f expected under Fe-stress would therefore reduce P$_{max-C}$. Indeed, P$_{max-C}$ was significantly correlated to physiological markers of Fe-stress, $\sigma_{PSII}$ ($\rho$ = -0.68, p << 0.01), F$_V$/F$_M$ ($\rho$ = 0.56, p = 0.02), and Si:NO$_3^-$ ($\rho$ = 0.60, p = 0.01). Additionally, meta transcriptomic analysis of diatom RNA revealed significant positive correlations (p < 0.05) between P$_{max-C}$ and the expression level of different PSI subunits (*psaE, psaL, psaM*), with correlation coefficients of $\rho$ = 0.71, 0.70, and 0.93, respectively. In contrast, there were no detected correlations between P$_{max-C}$ and Cyt b$_6$f, however *Cyt b559a*, a subunit of PSII, also demonstrated a strong positive

correlation with $P_{max-C}$ ($\rho = 0.76$, p < 0.05). While the precise functional roles of *Cyt b559a* are still not certain, previous studies have demonstrated its potential role in PSII assembly and photoprotective cyclic electron transport around PSII (Chiu and Chu, 2022). Finally, it is worth noting $K_{sat}$ and $P_{max-C}$ displayed a positive correlation ($\rho = 0.70$, p << 0.01).

In addition to non-linear electron transport, $\phi_{e:C}/n_{PSII}$ is also directly affected by the number of Chl energetically coupled to RCII ($1/n_{PSII}$). Directly measuring $1/n_{PSII}$ requires specialized $O_2$ flash yield instrumentation (Suggett *et al.*, 2009), which was unavailable for this study. Moreover, $O_2$ flash yield measurements may be challenging in natural plankton assemblages containing phytoplankton and bacteria. As an alternative, variability in $1/n_{PSII}$ between samples can be assessed based on

in-situ Chl concentrations, normalized to FRRF-derived proxies for [RCII] ($\propto F_o/\sigma_{PSII}$) following the approach of Oxborough et al. (2012). With a known instrument calibration factor, $K_a$, either provided by instrument manufacturers or determined independently by $O_2$ flash yield measurements, this approximation could be used to estimate the absolute value of $1/n_{PSII}$.

It is well established that Chl:RCII ($1/n_{PSII}$) ratios increase under low light, to maximize light absorption (Greenbaum and Mauzerall, 1991). In our measurements, the proxy for $1/n_{PSII}$ varied significantly between sample depths, with higher $1/n_{PSII}$ at the bottom of the euphotic zone compared to surface depths, confirmed by a t-test comparison of population means (p << 0.01). Iron limitation is also expected to increase Chl:RCII. Although iron limitation lowers total cellular Chl content, Chl is

more likely to be energetically coupled to RCII rather than PSI reaction centers (Greene et al., 1992). Accordingly, $1/n_{PSII}$ displayed a negative correlation with *Fea1* expression in surface samples ($\rho = 0.72$, p <0.05, n = 9), which we used as a proxy for iron limitation. We thus conclude that Fe-stress likely contributed to variability in $\phi_{e:C}/n_{PSII}$ in addition to influencing non-linear electron transport. The hyperbolic relationship between carbon fixation and electron transport was unaffected by $1/n_{PSII}$,

which was assumed to be constant for individual samples throughout the course of photosynthesis-irradiance experiments. However, this assumption may be violated under high light, due to photoinactivation of RCII (Campbell and Serôdio, 2020). A robust understanding of $\phi_{e:C}/n_{PSII}$ variability requires direct [RCII] measurements collected in parallel with $ETR_{PSII}$ and carbon fixation measurements.

**4.3.2 Comparison of $ETR_{PSII}$ and NCP**

As opposite end members of the productivity spectrum (Fig. 7), $ETR_{PSII}$ quantifies gross photochemical energy production, while NCP represents the net accumulation of photosynthetic carbon or oxygen remaining after accounting for all sources of mixed layer respiration. To directly compare $ETR_{PSII}$ and NCP, we converted $ETR_{PSII}$ from units of e$^-$ RCII$^{-1}$ s$^{-1}$ to mmol $O_2$ m$^{-2}$ d$^{-1}$ (Eq. 5) by assuming each

RCII was functionally coupled to 400 – 700 Chl pigments (Schuback et al., 2015; Kolber and Falkowski, 1993) and 4 charge separation events per gross $O_2$ evolved. The resulting $O_2$-based gross photochemistry values varied between 0 – 4000 mmol $O_2$ m$^{-2}$ d$^{-1}$ (Fig. 5c), coincident with the range of previously reported values for the CCS (Kranz et al., 2020). On average, NCP accounted for $17 \pm 8\%$

of gross photochemistry, indicating ~80% of oxygen produced at PSII by water splitting reactions was consumed within the mixed layer through autotrophic and heterotrophic respiration. In contrast to $\phi_{e:C}/n_{PSII}$, there was no significant differences in NCP:ETR$_{PSII}$ between Cape Blanco, Cape Mendocino, or offshore, suggesting minimal effects of nutrient limitation on decoupling between ETR$_{PSII}$ and NCP. Although NCP is constrained by gross photochemistry, NCP was greater than ETR$_{PSII}$ over 29% of the cruise track. This apparent contradiction can be explained by differences time-scales between instantaneous ETR$_{PSII}$ measurements, and NCP, which is integrated over $O_2$ residence times in the mixed layer (~1–2 weeks). Sustained net autotrophy can lead to accumulation of $O_2$ in the mixed layer, such that measured $O_2$ fluxes indicate high levels of NCP despite short-term decreases in ETR$_{PSII}$ (e.g. overnight).

Regardless of the large differences in integration time-scales and metabolic sources of decoupling, NCP showed strong coherence with ETR$_{PSII}$ (Fig 9). Direct comparison between continuous underway measurements of ETR$_{PSII}$ and NCP yielded a moderate positive correlation ($\rho = 0.43$, p $\ll$ 0.01). To account for some of the decoupling introduced by the strong diel dependence of ETR$_{PSII}$, we also compared ETR$_{PSII}$ and NCP measurements integrated over 24hr bins. This comparison indicated a much stronger relationship between ETR$_{PSII}$ and NCP ($\rho = 0.92$, p $\ll$ 0.01; Fig 9). For daily integrated time-scales, NCP linearly increased as a function of ETR$_{PSII}$ with a ~15% energy transfer efficiency and a predicted NCP of -0.55 mmol $O_2$ m$^{-2}$ when ETR$_{PSII}$ is zero. This efficiency estimate is within range of previous studies that have compared gross oxygen production and NCP using triple oxygen isotope and $O_2$/Ar methods (Howard et al., 2017; Haskell II et al., 2017), despite the differences in integration time scales between ETR$_{PSII}$ and the triple oxygen isotope method. Further, a sensitivity analysis found no significant changes in the derived energy transfer efficiency between ETR and NCP integrated over 24, 48, 72, and 96-hour bins. These results support the utility of FRRF to estimate gross oxygen productivity as an alternative to triple oxygen isotopes or other discrete methods, such as $H_2^{18}O$ tracer experiments.

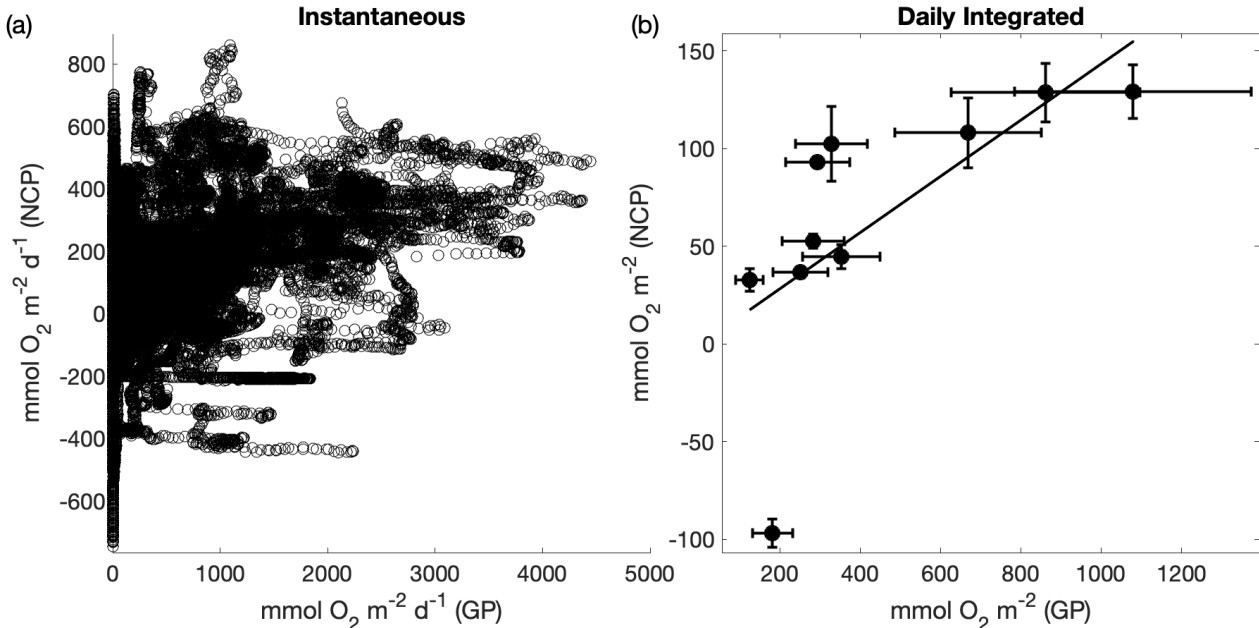


**Figure 9. Comparison between fluorescence-based gross photochemistry (GP) and NCP**. Panel (a) compares continuous underway $ETR_{PSII}$ and NCP measurements ($\rho = 0.43$, p << 0.01), panel (b) compares daily-integrated measurements of $ETR_{PSII}$ and NCP measurements ($\rho = 0.92$, p << 0.01). A line of best fit (y = 0.14*GP – 0.55) is drawn through the positive NCP data points. The error bars

represent the total uncertainty of each measurement determined by propagating uncertainty in each input variable. For GP, total uncertainty is represented as the uncertainty in the Chl:RCII. For NCP, uncertainty was quantified from uncertainty in the deep water supply ratio of $O_2:N_2O$, and modelled differences between $[N_2]$ and $[Ar]$. Assumption biases (e.g. no horizontal advection of $O_2$) also represents a potential large source of uncertainty but were not quantified.


Despite the inherent dependency of net oxygen production on gross oxygen production, the strength of the correlation between NCP and $ETR_{PSII}$ and the consistency of $ETR_{PSII}$:NCP across subregions is surprising given the multitude of methodological and physiological factors that can uncouple these rates (Fig 7). However, the derivations of NCP and GP both have similar dependencies on mixed layer Chl

concentration. To obtain FRRF-derived GP estimates in comparable units of mmol $O_2$ m$^{-2}$ d$^{-1}$, we multiplied in-situ $ETR_{PSII}$ by mixed layer Chl concentrations (Eq 5). While mixed layer Chl concentrations are not explicitly included in NCP calculations (Sect 2.6), biomass is expected to be a primary driver of bulk productivity. If Chl-normalized NCP is instead compared against GP expressed in units of mmol $O_2$ Chl$^{-1}$ d$^{-1}$, the correlation between 24h binned and instantaneous NCP and $ETR_{PSII}$

estimates decrease to $\rho = 0.22$ and 0.35, respectively. We therefore conclude that it remains challenging to derive gross and net carbon fluxes from FRRF measurements alone, but paired $ETR_{PSII}$ and Chl measurements can provide useful constraints for NCP estimates.

### 4.3.3 Final methodological considerations: spectral corrections

As discussed above, decoupling between $ETR_{PSII}$, C-fixation and NCP is affected by methodological factors, including differences in the time scale of different photosynthetic processes, and the different normalizations for various measured rates (e.g. per volume, Chl or RCII). Additionally, differences in the spectral characteristics of the in-situ light environment and instrument light sources may contribute further to decoupling between $ETR_{PSII}$, C-fixation, and NCP.


Phytoplankton exhibit variable light use efficiencies across the photosynthetically available wavelength spectrum, due to non-uniform pigment compositions and absorption spectra across assemblages. Consequently, photosynthetic measurements are wavelength-dependent (Kyewalyanga et al., 1997). As a result, differences in the spectral distribution of light between the FRRF and photosynthetron

incubator (used for $^{14}C$ uptake measurements) and the in-situ light environment could influence the stoichiometry we observed between $ETR_{PSII}$, C-fixation, and NCP.

In principal, spectral corrections, can be used to account for variability between instrument light sources and in-situ light environments to improve the inter-comparability of measurements (Schuback *et al.*,

2021; Tortell et al., 2023). These corrections require measurements of the spectral distribution of FRRF, photosynthetron incubator, and in-situ light, and a reconstruction of photosynthesis absorption spectra based on photosynthetic pigment concentrations, determined from HPLC analysis. These corrections rely on the assumption that absorption spectra of photosynthetic pigments accurately represent the action spectra of photosynthesis, which not always the case (Kyewalyanga et al., 1997). In our study,

we did not collect measurements of the spectral distribution of light in the euphotic zone, which would have likely varied significantly across depths and sampling sites (i.e. on-shore versus off-shore). For this reason, we are not able to spectrally correct in vitro measurements of $ETR_{PSII}$ and $^{14}C$-uptake to in-situ spectral fields for more direct comparison with NCP. As a result, spectral differences between instrument light sources and the in-situ light environment could contribute to some of the observed

variability between $ETR_{PSII}$, C-fixation and NCP in this study. As for the comparison of FRRF and $^{14}C$ data, spectral corrections would affect the absolute magnitude of $\phi_{e:C}/n_{PSII}$, but these corrections would yield a station-specific scalar and would not affect the hyperbolic relationship observed between $ETR_{PSII}$ and C-fixation.

The best approach to minimize the influence of spectral variability is to match the spectral properties of instrument light sources to ambient light fields. However, this remains challenging for high-resolution underway applications across varying spectral environments.

### 5 Conclusion

Consistent with previous observations, our results indicate a patchwork of Fe-stress within the coastal

upwelling waters of the California Current, with evidence for physiological Fe-stress within an upwelling filament near Cape Mendocino. Differences in iron availability between upwelling filaments

appear to be linked to bathymetric features that influence sediment loading, and variable micronutrient content of sub-surface upwelling source waters. Paired fluorescence- and $^{14}$C-based photosynthesis-irradiance measurements indicated strong connectivity between ETR$_{PSII}$ and carbon-fixation in Fe-replete phytoplankton, and greater decoupling in these rates for Fe-limited assemblages, with greater associated variability in $\phi_e : C/n_{PSII}$. Recently, there has been significant focus on understanding $\phi_e : C/n_{PSII}$ variability to expand FRRF-based GPP surveys (Hughes et al., 2018b). Our results suggest that nutrient replete phytoplankton are able maintain near constant $\phi_e : C/n_{PSII}$ under increasing excitation pressures due to their ability to efficiently transfer energy between PSI and PSII. Under these circumstances, NPQ$_{NSV}$ is not a good predictor of $\phi_e : C/n_{PSII}$. However, where nutrient limitation necessitates enhanced non-linear electron transport pathways to maintain energy balance between PSII and PSI, $\phi_e : C/n_{PSII}$ does scale with NPQ$_{NSV}$ and excess excitation pressure. With the accumulation of further data across a range of oceanographic conditions, it may be possible to derive more robust empirical relationships between NPQ$_{NSV}$ and $\phi_e : C/n_{PSII}$, which could be used to derive GPP in C-based units from FRRF measurements. In addition, our results show a general coherence between daily integrated GP, derived from ETR$_{PSII}$, and NCP measurements, suggesting that ETR$_{PSII}$ may have significant utility as an indicator of bulk primary productivity. We thus conclude that high-resolution, ship-board measurements hold significant potential to explore fine-scale variability in surface water primary productivity in complex coastal waters.

**Appendix A: Study region shelf width variability**

The shelf width was determined at each latitude as the distance between the 200m and 0m isobath calculated with the Haversine formula.

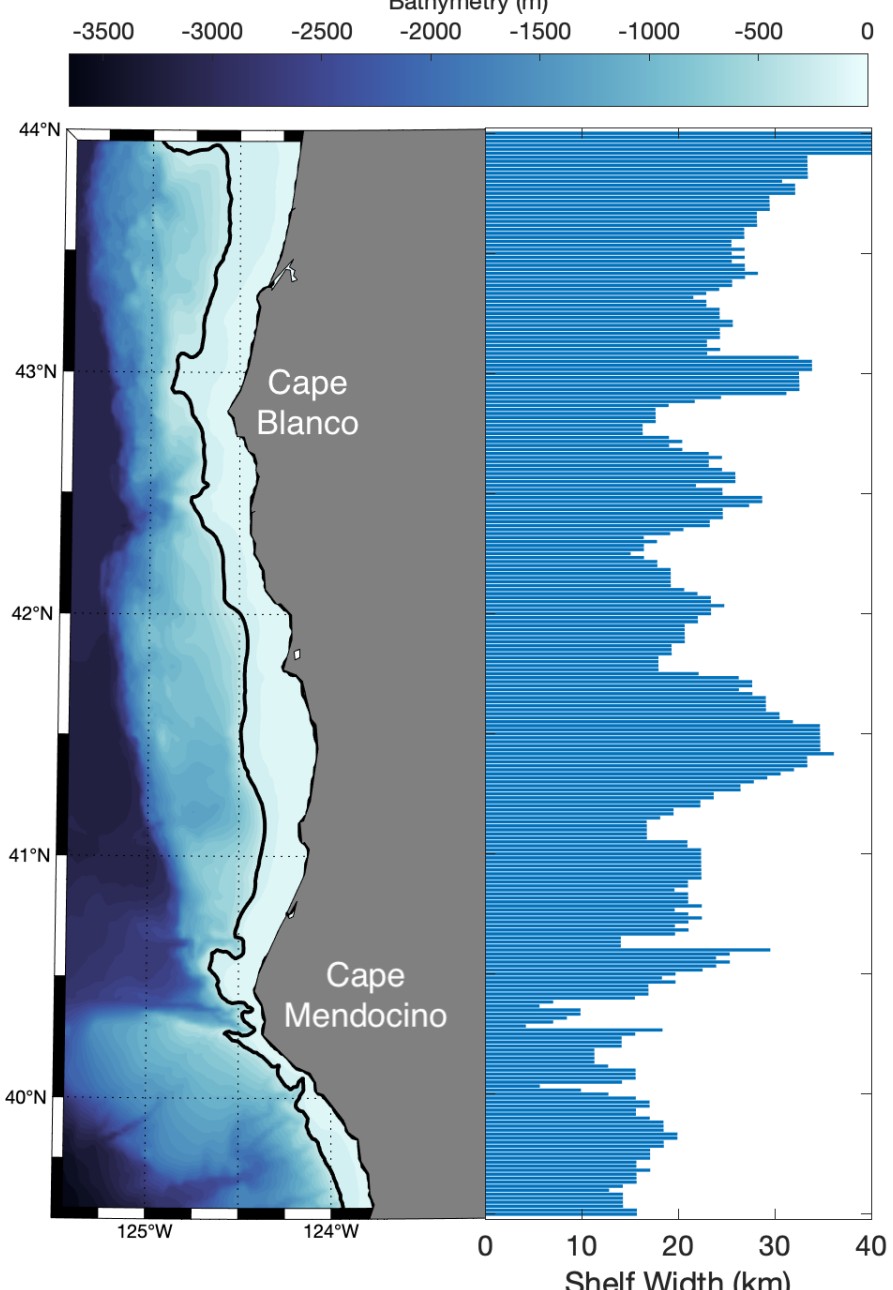

**Figure A1. Bathymetric map of the study region.** The black contour line indicates the 200m isobath. The horizontal bar graph demonstrates the shelf width at the aligned latitude.

## Appendix B: Photopigment and species distribution by sub-region

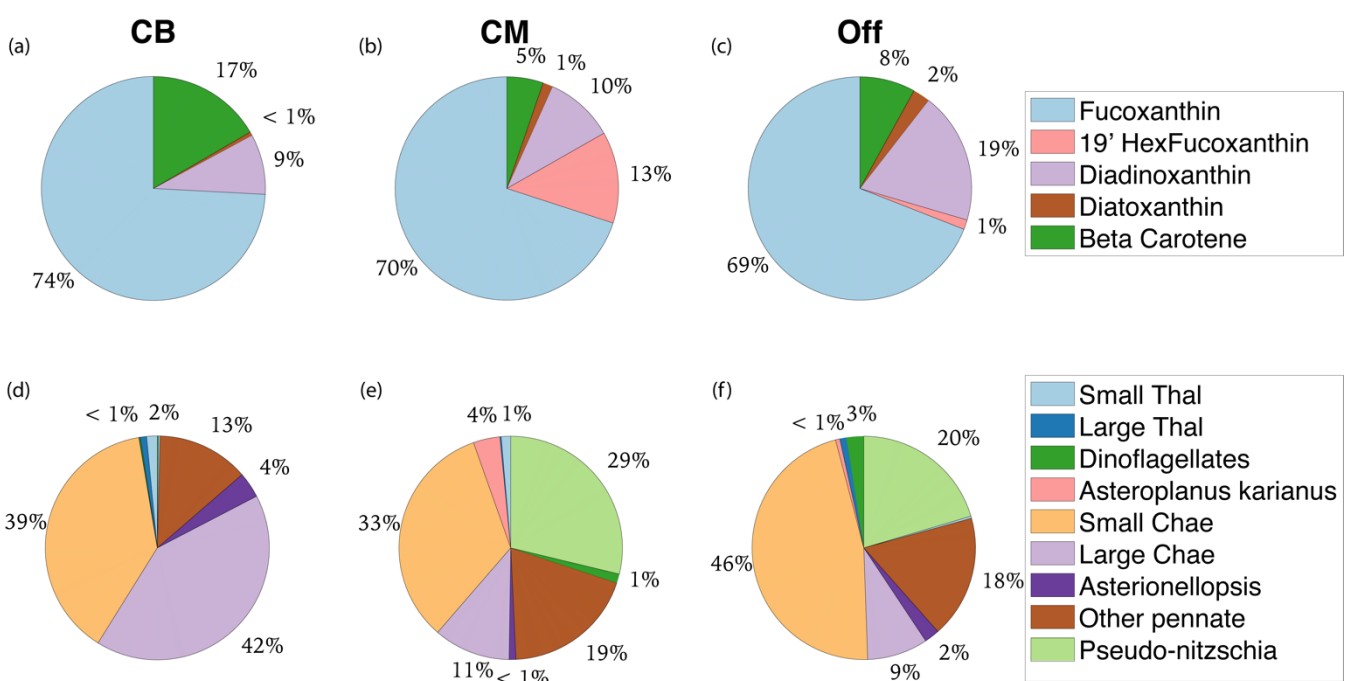

965 **Figure B1. Pigment and taxonomic composition of study area sub-regions**. a-c) display non-Chl pigment distribution by mass for Cape Blanco (CB), Cape Mendocino (CM), and Offshore (Off). d-f) display the taxonomic distribution of diatom and dinoflagellate groups visible for microscope counts.

## Appendix C: Normalized counts of diatom-specific *Fea1* transcripts

970

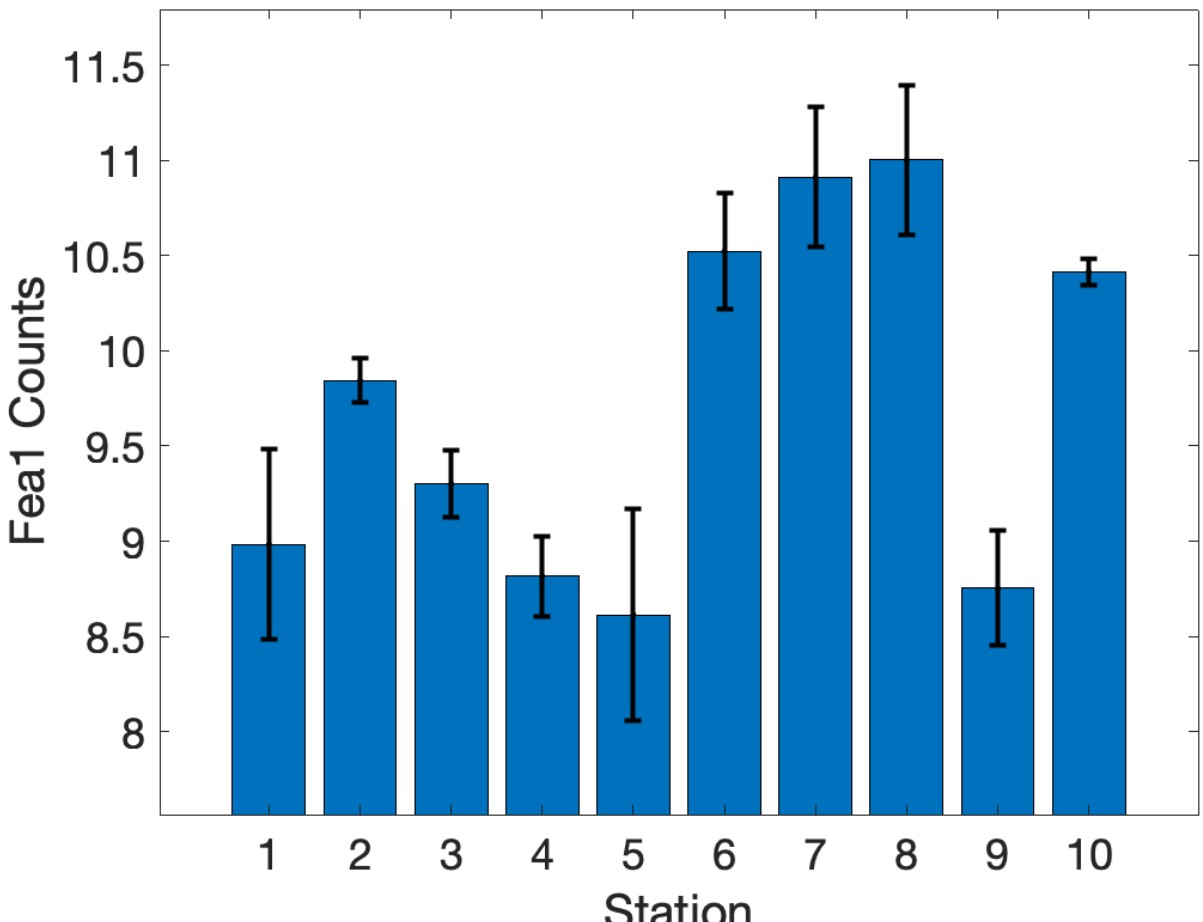

**Figure C1. Diatom-specific expression of Fe assimilation gene *Fea1*.** Bar height displays the mean of normalized *Fea1* counts mapping to Bacillariophyta by station. Normalized *Fea1* counts were calculated using DESeq2's median of ratios method in R (Love et al., 2014). Error bars display the standard deviation. Stations 1-4 were in close proximity of Cape Blanco, 6 and 10 were offshore while the remaining stations were in proximity of the Cape Mendocino upwelling filament.

## 7 Code and data availability

In-situ hydrographic (https://doi.pangaea.de/10.1594/PANGAEA.977780), dissolved gas concentration (https://doi.pangaea.de/10.1594/PANGAEA.977807), and phytoplankton photophysiology data (https://doi.pangaea.de/10.1594/PANGAEA.977630) collected during the PUPCYCLE II expedition are all publicly avaialable on PANGAEA, an Earth Science data repository. Dissolved Fe data have been submitted to https://www.bco-dmo.org/. DOI names are expected to be issued shortly. Satellite SST (11 $\mu$ daytime) and Chl data used to produce Fig 1 were downloaded from NASA Aqua MODIS platform

(https://oceancolor.gsfc.nasa.gov/l3/). Ancillary data required to derive $N_2'$ were sourced from several public databases. Wind speed and Ekman transport were taken from NOAA's Windspeed, Stress, Curl, Divergence, and Ekman Upwelling, Metop-C ASCAT, 0.25 degree, Global, Near Real Time, 2020-present, 1-Day Composite product. Sea level pressure was downloaded from the US Navy Global Environmental Model (NAVGEM) 0.5 degree, 2013-present Pressure MSL (https://coastwatch.pfeg.noaa.gov/erddap/griddap/erdNavgem05DPres.html). Modeled sea surface temperature and salinity products were downloaded from https://psl.noaa.gov/data/gridded/data.noaa.oisst.v2.highres.html and https://podaac.jpl.nasa.gov/dataset/SMAP_JPL_L3_SSS_CAP_8DAY-RUNNINGMEAN_V5, respectively. Model code used to calculate $N_2'$ and NCP calculations are available at https://github.com/rizett/O2N2_NCP_toolbox.

## 8 Author Contributions

Y.S. conceived the research plan, conducted FRRF and [14]C measurements, analysed data and wrote the manuscript with significant contribution from all co-authors. K.S. operated the PIGI, conducted all $N_2O$ measurements, and assisted with NCP computations and analysis. E.S. collected macronutrient and meta-transcriptomic data and assisted with meta-transcriptomic analysis. A.M. facilitated meta-transcriptomic data collection, assisted with meta-transcriptomic analysis, and co-organized the PUPCYCLE II expedition with C.T. Trace metal clean Fe samples were collected by C.T. and processed by R.T. Funding was secured by A.M., C.T., and P.T. Primary advisory support and manuscript editing was done by P.T.

## 9 Competing Interests

The authors declare they have no conflict of interest.

## 10 Acknowledgements

We would like to thank the crew and fellow scientists aboard the *R/V Sally Ride* who made this work possible and created a delightful working environment. Special thanks to Sacchinandan Pillai who provided HPLC data and mobilization assistance. We thank the Bundy lab at the University of Washington for the loan of their trace metal sampling van. We thank D. Patel from the University of North Carolina for performing microscope counts.

## 11 Financial Support

Funding for the PUPCYCLE II cruise was provided to AM from a National Science Foundation grant (OCE1751805). Iron analyses were funded through the Research Corporation for Science

Advancement's Cottrell Scholar Award #26844 to C.T. Trace metal sampling was conducted with a shared-use rosette maintained at Skidaway and purchased with NSF Award OCE-2015430.

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
