# Peer review of "Photosynthetic electron, carbon and oxygen fluxes within a mosaic of Fe limitation in the California Current Upwelling System"

_EGUsphere, 2024_

## Author Response (AR1)

**Response to Reviewer 1:**

We thank reviewer 1 for their positive feedback and edits, which strengthen this study. Our responses to reviewer 1's suggestions are outlined below in red. All edits are referenced by line number to the corresponding file of manuscript track changes.

L90: in the list of respiration pathways for NCP, shouldn't bacterial respiration as well as respiration by larger animals be added to this list? See also Figure 7.

We have amended the statement to 'community wide' respiration to cover respiration by respiration by bacteria and macrofauna. L91 in track changes file.

Figure 1: panels a and b are reversed in the caption.

Corrected, L160-162.

L151: was PAR measured or assumed? If it was measured, how was it measured?

Here, PAR is delivered by instrument LED lamps that provide continuous actinic light. PAR provided by the instrument was not directly measured during $ETR_{PSII}$ measurements, but the instrument LEDs were calibrated against a WALZ ULM-500 PAR meter prior to deployment. Note on calibration added L198

Eq 4: what is meant by "in situ PAR"? PAR averaged for the MLD? I'm struggling to make sense of the equation.

Yes, $PAR_{in-situ}$ refers to the mean PAR in the mixed layer. This equation closely follows that Eq. 3 in Domingues and Barbosa (2023), although some of the parameter names differ between our manuscript and their paper to maintain consistency within our manuscript. For example, here we denote the mixed layer depth as MLD, while Dominges and Barbosa use '$Z_m$'. Some text added for clarification (L234).

Eq 7: the superscript "B" – is it a designation (bio?) or an exponent? Can you make it a bit clearer by spelling it out?

Yes, B for Bio. Clarified text reads as follows: "Biological concentrations, indicated by the superscript, 'B', are derived by isolating and removing physical solubility effects from measured gas concentrations." (L334).

L262: inconsistency regarding letter k in the fraction: it's capitals here, but small-caps above (L253).

Corrected (L346)

L266: since it's the liquid fraction you're interested in, should it be "filtered through" rather than "onto"?

Corrected (L378).

Figure 2: could you indicate here what areas were used for the boxplots in Figure 3?

New panel added to Figure 3 to indicate data allocation between subregions.

[Figure]

**Figure 3. Variability in SST (a), salinity (b), NO$_3^-$ (c), and Chl (d) within the observed water masses offshore, at Cape Blanco, and Cape Mendocino**. The line inside each boxplot represents the median, while whiskers display the 75$^{th}$ percentile. Points outside the whiskers represent outliers. Panel (e) displays the spatial distribution of offshore (off), Cape Blanco, and Cape Mendocino (Mendo), data.

Note that although some 'Cape Blanco' data between 42.5-41.5N are at a relatively western longitude they are designated part of the coastal upwelling plume based on their low SST. Subregions were identified according to latitude and temperature criteria. Samples were considered part of an upwelling group if they had SST < 12 °C. If upwelling samples were north of 41N they were considered part of the Cape Blanco plume. If south of 41N, they were designated Cape Mendocino samples. All other samples were grouped in the offshore category. Criteria were determined by studying Figure 1a.

L365-368: These statements could distinguish a bit more between the capes – not all statements are true for both capes (Figure 4).

In reviewing Figure 4, there is evidence of diel cycles in Fv/Fm, $\sigma_{PSII}$, and NPQ$_{NSV}$. However, it is true that the magnitude of apparent diel cycles varies between Cape Blanco and Cape Mendocino. To acknowledge that, we have added the statement, 'the magnitude of diel variability in $F_V/F_M$, $\sigma_{PSII}$, and NPQ$_{NSV}$ signals displayed significant variability between subregions, as discussed below' (L517). In-depth analysis of differences in photo-physiology between the two capes is discussed in the following paragraphs and Table 1.

Additionally, there is some convolution of the diel signals since the ship typically stayed close to the shelf during the day and transited offshore overnight. We have already acknowledged this in L535-544; 'We note, however, that there is potential for some convolution of temporal and spatial variability, as the ship spent more time offshore in the night, and on-shore during the daytime. It is thus possible, that some of the diel cycling partially reflects different photo-physiological signals between coastal and offshore waters.'

L378/79: this statement is not true for alpha at Cape Blanco – this whole section could be either a bit more refined, or throw in a few more "overall"s to indicate that what is said is not always true but a general trend.

Threw in some generalizing terms: 'Generally' (L512), 'Overall' (L529)

L380: is beta shown anywhere?

Didn't end up including beta in our plots but included beta in Table 1 now.

L387: "Elevated values…" – this is only true for Cape Blanco. You said previously that both capes are upwelling, but to different degrees; maybe start early with a clear distinction between the two capes, but "upwelling" is not one of them, that's true for both (see also L389).

Updated the text to reflect the general trends and relationships reported in Table 1 between upwelling conditions and photo-physiology, while still setting up the reader to understand how/why photo-physiology can still differs between two upwelling areas.

L547-553:

In general, $F_V/F_M$, P$_{max}$, E$_k$ and $\alpha$ displayed positive relationships with upwelling indicators, i.e. salinity, nitrate, and decreased sea surface temperature (Table 1), suggesting that vertical transport of nutrient-rich water to the surface supported high photochemical yields. In contrast, signs of upwelling were associated with decreased $\sigma_{PSII}$ and NPQ$_{NSV}$. However, despite general trends between photo-physiological parameters and upwelling, there were significant differences in photo-physiological properties between the Cape Blanco and Cape Mendocino upwelling plumes. At Cape Blanco…

L416/17: Is this the "in situ PAR" that confused me above? If so, make sure to call it that in the Figure.

Yes, added to figure 5a legend and caption.

L425-429: refer to Figure 5 here?

Done (L588).

L440-444: maybe add a word here about the uncertainties (due to nitrification in the surface layer) that were mentioned in the methods? What order of magnitude are they?

Added an estimate of error due to euphotic zone nitrification and added some details to the methods describing how this error was determined.

L348-358:

"We note that several recent studies have observed nitrification within the euphotic zone, challenging the assumption that $N_2O$ production is limited to subsurface waters (Grundle, Juniper and Giesbrecht, 2013; Smith *et al.*, 2014), and potentially leading to overestimates in our vertical mixing-corrected NCP estimates. Previous observations in the CCS reported a range of depth-integrated mixed layer nitrification rates between $0.3 - 2$ mmol $NH_4^+$ $m^{-2}$ $d^{-1}$, resulting in consumption of $0.6 - 4$ mmol $O_2$ $m^{-2}$ $d^{-1}$ (Stephens *et al.*, 2020). Following the approach of Izett et al. (2018) we used a range of realistic $N_2O:O_2$ stoichiometries to estimate potential upper and lower bounds of mixed layer $N_2O$ production. We determined mixed layer $N_2O$ production likely ranged between $0.09 - 0.23$ $\mu$mol $N_2O$ $m^{-2}$ $d^{-1}$, which would yield offsets in our final NCP estimates between 1.2 and 3.3 mmol $O_2$ $m^{-2}$ $d^{-1}$. Total uncertainty due to sources of error in other derived parameters was determined by following Izett (2021).

L627 – 629:

"Uncertainty in vertical-mixing corrected NCP due potential mixed layer nitrification (see sect. 2.6) represented between $1.5 - 4.2\%$ of our mean corrected NCP value."

L451: "x+/-y" wants to be filled with numbers

Yikes! Amended with the values.

L641-642: "On average, the bottom depth of the euphotic zone was $14 \pm 12$ meters deeper than the bottom depth of the mixed layer"

Figure 7 and its caption: Do you mean "spatial and time scales"? Only the time scales are mentioned in the Figure. Also, I can't see pathway 7 anywhere in the Figure

Removed 'spatial' and added a 7 symbol in the NCP panel.

L633: Did you mean to refer to Figure 6 here (rather than Figure 8)?

Removed the reference to figure 8, which does not belong here.

L653: "… with low Cmax" – this should be "…with low Pmax-C", no?

Corrected.

Figure 8 and its caption: maybe make it clearer in the caption that blue and red also refer to different Fe-limitation regimes, or make a second panel where the dots are coloured by some measure of Fe limitation (Fv/Fm, sigma,… something like that)? This is a useful distinction, no? Also, Vmax (mentioned in the caption) is not shown, nor do I know what it would be based on what has been discussed?

$V_{max}$ in caption corrected to $P_{max-C}$. Added some text to the Figure 8 caption.

L664: mention of Vmax again – should this be Ksat? It's getting a bit confusing in this paragraph… if there is both a Vmax and a Ksat, then maybe a table of all the variables and their meaning would be useful?

Apologies for the confusion between Vmax, Pmax-C and Cmax. Pmax-C is the maximum Chl-normalized carbon fixation rate. The parameter pmax-C is comparable with Vmax in traditional Micahelis-Menten terminology. However, the maximum carbon fixation rate, whether expressed as a function of light, as in a PI curve, or as a function of ETR is nearly the same (no difference in the maximum measured carbon fixation rate, but the derived value changes very slightly between models). When writing this manuscript, there were many iterations where we tested different parameter names for Pmax-C, and some of the old names, Cmax, Vmax, etc, were not caught during editing. Thank you for your careful attention to this detail! Section 4.3.1 has been carefully reviewed now to catch all mentions of Cmax/Vmax/or Pmax-C and make sure they all read $P_{max-C}$.

L696/70: I would have thought that *Cyt b559a* would correlate with NPQ, and therefore expected a negative correlation with Pmax-C. Does this deserve a bit more discussion?

I would agree that cyt b559a would be expected to positively correlate with NPQ, however I would disagree that Pmax should be negatively correlated with NPQ. Samples with high and low Pmax-C both expressed high levels of NPQ. One of the main findings of this section is that in some cases, carbon fixation cannot be predicted from NPQ. The text has been modified as follows:

While the precise functional roles of *Cyt b559a* are still not certain, previous studies have demonstrated its potential role in photoprotective cyclic electron transport around PSII and PSII assembly (Chiu and Chu, 2022). L1004-1007.

L678: Vmax again… and also, a question that comes up reading this paragraph and the preceding ones: How do you decide whether to discuss Vmax (Ksat) rather than Pmax-C? Is there a scheme to this? They seem somewhat interchangeable to me at this point. When

you don't mention one or the other, does it mean the correlation would have been insignificant? Again, maybe a word or two on this would be useful.

Text around here has changed quite a bit so that text is gone now (see following comment). However, the point about interchangeable parameters is still a good one.

The text has been cleaned up so only Pmax-C is mentioned throughout. $K_{sat}$ and Pmax-C are not interchangeable. Rather, the ksat and Pmax-C terms discussed here are comparable with the Ksat and Vmax terms in a Michealis-Menten equation.

Discussion is mostly limited to Pmax-C for simplicity and because this parameter is more representative of photosynthetic enzymes concentrations and kinetics, which provides mechanistic insights into photosynthetic functioning and are useful to understand the observed variability in $\phi_{e:C}/n_{PSII}$. It's worth noting that $K_{sat}$ and Pmax-C are positively correlated ($\rho$ = 0.70, p << 0.01) L1006-1007.

L678-80: This sentence is either missing a final section, or the "and" after the comma needs to go; also, please specify whether this increased coupling of Chl to RCII is specific to iron limitation or generally true.

Lots of new details added to the discussion of Chl:RCII and its influence on our comparison of carbon fixation and electron transport rates 1009-1033:

"In addition to non-linear electron transport, $\phi_{e:C}/n_{PSII}$ is also directly affected by the number of Chl energetically coupled to RCII ($1/n_{PSII}$). Directly measuring $1/n_{PSII}$ requires specialized $O_2$ flash yield instrumentation (Suggett *et al.*, 2009), which was unavailable for this study. However, in-situ Chl concentrations, normalized to FRRF-derived proxies for [RCII] ($\propto$ $F_0/\sigma_{PSII}$) following the approach of Oxborough et al. (2012), can be used to examine variability in $1/n_{PSII}$ between samples. With a known instrument calibration factor, $K_a$, either provided by instrument manufacturers or determined independently by $O_2$ flash yield measurements, this approximation could be used to estimate the absolute value of $1/n_{PSII}$.

It is well established that Chl:RCII ($1/n_{PSII}$) ratios increase under low light, to maximize light absorption (Greenbaum and Mauzerall, 1991). In our measurements, the proxy for $1/n_{PSII}$ varied significantly between sample depths, with higher $1/n_{PSII}$ at the bottom of the euphotic zone compared to surface depths, confirmed by a t-test comparison of population means (p << 0.01). Iron limitation is also expected to increase Chl:RCII. Although iron limitation lowers total cellular Chl content, Chl is more likely to be energetically coupled to RCII rather than PSI reaction centers (Greene *et al.*, 1992). Accordingly, $1/n_{PSII}$ displayed a negative correlation with *Fea1* expression in surface samples ($\rho$ = -0.72, p <0.05, n = 9), which we used as a proxy for iron limitation. We thus conclude that Fe-stress likely contributed to variability in $\phi_{e:C}/n_{PSII}$ in addition to influencing non-linear electron transport. The hyperbolic relationship between carbon fixation and electron transport was unaffected by $1/n_{PSII}$, which was assumed to be constant for individual samples throughout the course of photosynthesis-irradiance experiments (Appendix 4). However, this assumption may be violated under high light, due to photoinactivation of RCII (Campbell and Serôdio, 2020). A robust understanding of $\phi_{e:C}/n_{PSII}$

variability requires direct [RCII] measurements collected in parallel with $ETR_{PSII}$ and carbon fixation measurements."

L681: "increases in"? or "increased"? This whole paragraph could be a bit clearer - what does the literature say, what was measured here, what needs to be assumed, how does it all come together?

This text has been replaced with the paragraphs above.

L704: Can you insert the actual ranges that have been observed by others?

Kranz et al. (2020) observed GPP rates of 0 – 4000 mmol C $m^{-2}$ $d^{-1}$, 0 – 4000 mmol $O_2$ $m^{-2}$ $d^{-1}$ from $O_2$/Ar- and FRRF- based approaches. This range is coincident with that observed in this study (L1042).

Paragraph starting L698: The observed relationship between FRRf-derived GPP and NCP is a significant and surprising result (maybe you disagree, but then please lay out more clearly why it's not surprising). Maybe a bit more discussion on why this relationship with NCP works here would not be amiss. Where/when do we expect this to work elsewhere (and where/when may it fall over)? What needs to be investigated next? Does it have anything to do with iron at all?

Agreed! This is a significant and perhaps surprising result, due to the large differences in integration time scales of GPP and NCP, and the potential for different metabolic and ecological processes to decouple these rates.
We initially hypothesized that ETR:NCP would vary with ETR:C-fixation, and that factors, like Fe availability, that drive decoupling between ETR and GPP would similarly decouple ETR and NCP. However, we did not observe significant differences in ETR:NCP between Cape Blanco and Cape Mendocino as would be expected if Fe was playing a key role in ETR:NCP variability. Rather, the relationship between ETR:NCP was fairly consistent across the study region. We offer an alternative explanation for the surprising consistency in the ETR:NCP relationship below (L1136-1147):

Despite the inherent dependency of net oxygen production on photosynthetic electron transport rates, the strong correlation between NCP and $ETR_{PSII}$ and the consistency of $ETR_{PSII}$:NCP across our entire study regions is surprising given the large number of methodological and physiological factors that can significantly uncouple these rates (Fig 7). However, NCP and GP estimates both have similar dependencies on mixed layer Chl concentration. To obtain FRRF-derived GP estimates in comparable units of mmol $O_2$ $m^{-2}$ $d^{-1}$, we multiplied in-situ $ETR_{PSII}$ by mixed layer Chl concentrations (Eq 5). Although mixed layer Chl concentrations are not explicitly included in NCP calculations (Sect 2.6), biomass is expected to be a primary driver of bulk productivity. Indeed, when we compared Chl-normalized $ETR_{PSII}$ and NCP estimates, we found a much weaker relationship between with correlation coefficients decreasing to 0.22 and 0.35 for 24h binned and instantaneous measurements, respectively. We therefore conclude that it remains challenging to derive gross and net carbon fluxes from FRRF measurements alone, but paired $ETR_{PSII}$ and Chl measurements can provide useful constraints for NCP estimates.

Figure 9 and caption: Does the correlation for panel b (rho = 0,92,L715) include the negative data point? How should one think about the negative data point, can it be explained, could it have to do with upwelling/downwelling? And what about the intercept for the line of best fit (-0.55), how to think about that? That it's very close to zero and that's good, or does it have some other meaning that's worth spelling out? Also, the final sentence of the figure caption is a repeat, except for the r2. Regarding panel A of the figure: could it be improved by plotting the points as a heat map, to show where most of the data points fall?

The correlation does include the negative data point. This point corresponds to measurements on June 2. On this day NCP reached its minima. This point corresponded with high SST indicative of offshore waters outside the upwelling plume (see Fig 5b).

The y-intercept being close to zero is a positive indication that our 24h binned NCP and ETR comparison are reasonable, however this y-intercept should not be taken as an absolute. As we can see from the highly negative point in Fig 9b, very negative NCP values are possible in net heterotrophic waters, while ETR values can never be net negative.

Repeat sentence removed from Fig 9 caption

[Figure]

A heat map of the data density indicates that most of the data is concentrated around [ETR = 0, ncp = 100]. This makes sense since ~half the ETR data should = 0 overnight and the mean ncp was about 80 mmol O2 m$^{-2}$ s$^{-1}$. Measurements collected during the day (ETR > 0) appear evenly spread. Since the point that ETR must equal 0 overnight is made a few times in the text, we have decided not to change the manuscript in the figure to the heat map, but hope it is of interest for Reviewer 1.

L739: Regarding "fine-scale variability": what scales are being resolved when we go to 24h binning?

This is an excellent point! A primary motivation for using FRRF and O2/N2 is the ability to capture fine-scale variability. That fine-scale variability is of course smeared if a 24h bin is applied. The spatial 'smear' will depend on the boat's movement throughout the day. In our case, the boat stayed on-station during the day for full-scale morning and afternoon sampling programs and transited to the following station overnight.

Here each 24h binned ETR measurement roughly represents the mean photochemical activity at each station over 24 hours, since ETR is zero overnight when the ship was transiting. The interpretation is more complicated for NCP which captures variability in O2/N2' overnight due to variable respiration rates as the ship transited through different water masses.

Binning obviously negates the critical advantages of high frequency measurement systems like the FRRF and PIGI. However, we applied this approach to purposefully smear the temporal resolution of ETR to approximate the time-scales represented by O2/N2'-based NCP measurements and provide a more just comparison. This approach is far from perfect, but we found it useful to illustrate how trying to account for differences in time-scales can improve the correlation between productivity metrics.

**Response to Reviewer 2**

We similarly thank Reviewer 2 for their positive feedback, expert insights, and suggestions for clarification and deeper analysis. We have aimed to incorporate as many of their suggestions as possible. Unfortunately, not all the datasets they have requested for deeper analysis are available. Where further data analysis is not possible, we strive to incorporate the additional considerations Reviewer 2 suggests in our discussion. Again, Reviewer 2's suggestions are listed below in black with our replies in red. Line references in our replies correspond to the lines in the updated manuscript with track changes.

[Suggestions]

A few mentions of phytoplankton taxonomy

1. During your sampling, diatoms, particularly *Cheatoceros*, dominated the phytoplankton communities in both upwelling areas. Undoubtedly, diatoms were most abundant in your samples; however, you might also observe other diatoms, dinoflagellates and/or small taxa. You conducted light microscopy to look at the phytoplankton taxonomy of your samples. If your microscopy data was sufficiently reliable, subsection 4.2 can be improved as Taguchi (1976), Finkel (2000), and Suggett et al. (2009) reported that different taxa showed unique photosynthetic performances even both carbon and fluorescence.

Our taxonomic analysis primarily relied on chemotaxonomic analysis of pigment data which does not differentiate between diatom taxa. Microscopic analysis was limited to surface samples and was useful to validate our chemotaxonomic analysis and identify general trends between stations, i.e. presence of Pseudo-Nitszchia at Cape Mendocino, but not Cape Blanco. However, this analysis is pretty biased towards larger more recognizable cells and was used more qualitatively. This data is likely not sufficient to apply robust statistical analyses to evaluate taxonomically driven variability in photo-physiology. This is unfortunate since, as you say, it is clear from the literature that different taxa express different optical phenotypes influencing Chlorophyll fluorescence signatures.

In a field study, you could argue that variability in these optical phenotypes still ultimately derives from the environmental conditions that select for different phytoplankton taxa.

More discussion on Carbon/nitrogen ratio

1. I agree Fe availability modifies the photosynthesis machinery such as PSI: PSII and Cytochrome: PSII, which decouples the electron: carbon conversion. However, photosynthetic energy can be allocated to both carbon and nitrogen et cetera, which could also decouple the conversion factor. You only discuss C: N ratios compared with the Redfield ratio. Discussion on the relative allocation to nitrogen from C: N ratios can improve your discussion on the decoupling electron: carbon.

Agreed, N-assimilation is a decoupling pathway that is not directly triggered by excess energy and could therefore help explain why NPQ and e:C are not always correlated. Further, the source of N is also likely to affect e:C as growth on more reduced forms (i.e. $NH_4$) will divert less e from carbon fixation. However, N-assimilation is also negatively impacted by Fe-limitation as both nitrate and nitrite reductase use Fe as cofactors. Allen et al. (2008) demonstrated these enzymes were downregulated under Fe limitation. While previous studies of diatoms have found either no change in cellular N:P or decreases in N:P in response to Fe limitation (Price 2005; La Roche et al., 2003). As a result, it is not clear how C:N may have varied between our Fe rich and Fe stressed study sites, or how Fe limitation may have influenced e:C decoupling through nitrate assimilation. This is an important point to resolve for the community moving forward. Unfortunately, without the accompanying data, we feel this topic is somewhat out of the scope of our paper.

Derivation of qP or qL

Your discussion on low Fe effects on FRRf and carbon fixation was on non-linear electron transport due to an imbalance between photosystems. I understand you well discuss the reduction of PSI, inferred from TPMs (or relative gene expression) of the PSI-related genes. In addition, I would suggest you calculate 1-qP or 1-qL as the state of open/closed PSII from your FRRf data.

Yes, qP (fraction of open RCII) plays an important role in determining $ETR_{PSII}$, and is actually included in our $ETR_{PSII}$ equation (Eq.1) where it is shown under its other notation, F'q/F'v (see Tortell, Suggett and Schuback 2023 for a list of synonymous FRRF nomenclature). While we did discuss potential Fe effects on F'q/F'v (L808-809), we did not take the next steps in actually analyzing our F'q/F'v data. Following Reviewer 2's suggestion, we have now actually examined qP along the cruise transect (see below).

[Figure]

The figure above displays the parameter 1-qP. qP was measured at light levels ranging from 0-850 uE during photosynthesis-irradiance curves. To estimate the fraction of closed RCII, in-situ, we applied the same approach we used to determine underway NPQ, where 1-qP was plotted against PAR, and fit with an exponential curve (92% of model fits had R2 > 0.9). In-situ 1-qP was then estimated by inputting in-situ PAR into the resulting model equation. Our results demonstrate strong diel patterns due to the immediate dependence of this approach on PAR, but also highlight that the fraction of RCII closure during peak midday irradiance was typically lower for Cape Blanco samples (5/30 – 06/04) compared to Cape Mendocino (06/06 – 06/10), with the exception of data collected on 06/02, which is when we transited offshore (note, this is also where we recorded our lowest NCP measurement).

To minimize confusion and avoid adding a new parameter name in the text, we are opting not to use the qP terminology and remain consistent with F'q/F'v. We have added analysis of F'q/F'v to Table 1, and added a note to our discussion section (L812-814):

'Indeed, $F_q'/F_v'$ measured during underway Photosynthesis-Irradiance curves and mapped onto in-situ irradiances, demonstrated that $F_q'/F_v'$ was higher around Cape Blanco compared to Cape Mendocino (Table 1)'

We also added some text to the methods (Section 2.4) to explain how we evaluated in-situ F'q/F'v. (L273-276)

How much RNA data did you throw away?

Please see the table below contributed by coauthor Emily Speciale detailing our sequencing stats.

| Station | Rep | Total Reads | # Reads Mapped | % Reads Mapped | # Reads Taxonomically Annotated | % Reads Taxonomically Annotated | # Protist Group | % Protist Group | # Diatom | % Diatom |
|---|---|---|---|---|---|---|---|---|---|---|
| 1 | A | 20919052 | 15566516 | 74.41 | 12163291 | 58.14 | 8529884 | 40.78 | 4462784 | 21.33 |
|  | B | 20605487 | 14828573 | 71.96 | 11629423 | 56.44 | 8495042 | 41.23 | 5035031 | 24.44 |
| 2 | A | 20370660 | 16757515 | 82.26 | 12441421 | 61.08 | 9060238 | 44.48 | 3337286 | 16.38 |
|  | B | 33411917 | 23851899 | 71.39 | 17576726 | 52.61 | 3878516 | 11.61 | 1577339 | 4.72 |
|  | C | 20451261 | 16318259 | 79.79 | 13878818 | 67.86 | 6014888 | 29.41 | 3218519 | 15.74 |
| 3 | A | 31854033 | 24637176 | 77.34 | 17205795 | 54.01 | 8931444 | 28.04 | 1585515 | 4.98 |
|  | B | 23102631 | 18257440 | 79.03 | 12937513 | 56.00 | 7632094 | 33.04 | 1385720 | 6.00 |
|  | C | 22184188 | 9543967 | 43.02 | 5778682 | 26.05 | 1516238 | 6.83 | 601494 | 2.71 |
| 4 | A | 31244147 | 22594019 | 72.31 | 14833206 | 47.48 | 8988587 | 28.77 | 3345948 | 10.71 |
|  | B | 17121856 | 13681984 | 79.91 | 9435876 | 55.11 | 6813257 | 39.79 | 2557063 | 14.93 |
|  | C | 21784053 | 16405679 | 75.31 | 12470858 | 57.25 | 9922004 | 45.55 | 5794848 | 26.60 |
| 5 | A | 24551450 | 19993489 | 81.44 | 16862370 | 68.68 | 15686918 | 63.89 | 3354493 | 13.66 |
|  | B | 20633218 | 15952493 | 77.31 | 13714182 | 66.47 | 12762334 | 61.85 | 3482216 | 16.88 |
|  | C | 13037287 | 5936298 | 45.53 | 4437254 | 34.04 | 632351 | 4.85 | 253280 | 1.94 |
| 6 | A | 18605896 | 14369514 | 77.23 | 11475963 | 61.68 | 8299338 | 44.61 | 2787489 | 14.98 |
|  | B | 20757774 | 7441701 | 35.85 | 3668990 | 17.68 | 1359173 | 6.55 | 441358 | 2.13 |
|  | C | 22992753 | 18563929 | 80.74 | 14898492 | 64.80 | 10666864 | 46.39 | 4200265 | 18.27 |
| 7 | A | 13046360 | 10221886 | 78.35 | 6465869 | 49.56 | 4729515 | 36.25 | 819835 | 6.28 |
|  | B | 22985192 | 18162597 | 79.02 | 13223369 | 57.53 | 8650540 | 37.64 | 1678663 | 7.30 |
|  | C | 17752669 | 8665680 | 48.81 | 4882063 | 27.50 | 2656758 | 14.97 | 840107 | 4.73 |
| 8 | A | 28417725 | 21660636 | 76.22 | 12402752 | 43.64 | 8187261 | 28.81 | 1294051 | 4.55 |

| | | | | | | | | | | |
|---|---|---|---|---|---|---|---|---|---|---|
| | B | 18902138 | 15344332 | 81.18 | 9352687 | 49.48 | 5208950 | 27.56 | 901508 | 4.77 |
| | C | 18139851 | 12084503 | 66.62 | 9296668 | 51.25 | 6835677 | 37.68 | 1183001 | 6.52 |
| 9 | A | 22252339 | 17619164 | 79.18 | 14355113 | 64.51 | 13217881 | 59.40 | 1699119 | 7.64 |
| | B | 22274606 | 17295073 | 77.64 | 14039185 | 63.03 | 12776475 | 57.36 | 1600294 | 7.18 |
| | C | 18592697 | 15045958 | 80.92 | 11457739 | 61.62 | 10642495 | 57.24 | 1281205 | 6.89 |
| 10 | A | 19985953 | 17084390 | 85.48 | 11958397 | 59.83 | 8132358 | 40.69 | 1241650 | 6.21 |
| | B | 14932416 | 12288932 | 82.30 | 8724712 | 58.43 | 5509449 | 36.90 | 791170 | 5.30 |
| | C | 19818129 | 11020988 | 55.61 | 6792417 | 34.27 | 2580105 | 13.02 | 628061 | 3.17 |
| | | | | | | | | | | |
| SUM | | 620727738.00 | 451194590.00 | 72.69 | 328359831.00 | 52.90 | 218316629.60 | 35.17 | 61379310.90 | 9.89 |
| AVG | | 21404404.76 | 15558434.14 | 72.28 | 11322752.79 | 52.62 | 7528159.64 | 35.35 | 21165297.96 | 9.90 |
| STD. DEV. | | 4905885.70 | 4678756.37 | 13.18 | 3827375.65 | 13.20 | 3751501.90 | 16.77 | 1490867.62 | 6.91 |

I agree your transcriptomics indeed deepened your discussion. I have two technical questions:

(1) Have you compared with other de novo assemblers than rnaSpades or CD-HIT-EST? Any assembling performance data? Did you pre-process raw RNA sequences?

Coauthors Adrian Marchetti and Emily Speciale are part of an OCB Meta-Eukmoics working group that is doing an intercomparison of metatranscriptomic methods for microbial communities. Based on the work we've done thus far, there are no clear trends between the performance of different de novo assemblers (whether that be rnaSpades, Megahit, Trinity, etc.). Thus, we chose to use rnaSPAdes and CD-HIT-EST due to our familiarity with the softwares and their efficiency. We did pre-process raw reads using TrimGalore and FastQC.

(2) You extracted diatom reads/contigs for further downstream analysis. However, transcriptomic data might be unbiased and might have various reads/contigs from various organisms. What proportion did diatom data contribute to the total sequences? How much RNA data did you throw when only focusing on diatoms?

See the table above for information regarding the proportion of transcriptomic data.

We chose to focus on diatoms because we believe they dominant functional group among the photoautotrophs in this ecosystem and there is rich literature offering sufficient references to

interpret diatom transcriptomic data. There are definitely substantial non-diatom organisms in our metatranscriptomics data that map to non-autotrophic organisms. When trying to look at broad expression trends, trophic mode can be hard to filter for, so we filtered for Bacillariophyta (aka diatom) mapped reads to represent the dominant phytoplankton response to various environmental conditions.

[Minor issues]

L110: Have you validated your nitrate sensor with your autoanalyzer outputs?

The sensor was recently recalibrated by Seabird prior to the cruise. Immediately prior to the cruise and during the cruise itself we validated the sensor against standards of sodium nitrate dissolved in MQ. We have added a note of this to the methods section (L145-146).

L139: How did you dissimilate/separate QA and QB reoxidation from your FRRf data? You can separate QA and QB if using an algal isolate, but was it appropriately separated in natural communities even with various Fe availabilities?

The FRRF method we applied is a single-turnover method which relies on the assumption that our full sequence of sub-saturating excitation pulses and following relaxation sequence leads to a single reduction and oxidation event of the collective (mixed assemblage) Qa pool. This assumption is based on the timing of the excitation sequence (127 sub-saturating pulses delivered over 250 us) which outpaces the reoxidation of $Qb^=$ -> $PQH_2$ -> Cyt $b_6f$ which occurs over 4-5 ms and represents one of the rate limiting steps of photosynthetic light reactions. Multiple turnover methods with longer excitation time-scales (200 – 10,000 ms; e.g. PAM fluorometers) do record fluorescence signals associated with Qb reoxidation (see figure below).

[Figure]

Multiple Turnover (PAM)

[Figure]

Single Turnover (FRRf & STAF)

Image credit: Chelsea Instruments. The multiple turnover shows a temporary pause in the rise of fluorescence when Qa is saturated. As Qa is reoxidized, fluorescence continues to

increase until the entire electron transport chain is saturated. This is part of the reason Fv/Fm measured by PAM fluorometry is higher than FRRF.

L203: I understand NPQ fitting is quite tricky. Serôio and Lavaud (2011) thus discussed model performance on NPQ-E fittings. How were your single-component exponential fittings? Any stats outputs?

Stats added (L270): 'Out of 91 curve fits, 95 % had $R^2 > 0.90$ and 87% had $R^2 > 0.95$.

A sample NPQ-PAR curve is included below.

[Figure]

L206: Why did you choose 46% light depth? You surely collected the same light depths at each station? Or simply light intensities at ~50 m almost similar at each station?

A primary objective of this cruise campaign was to collect measurements during a large scale Fe enrichment (+Fe) and depletion (+DFB) incubations. Light depths were sampled to mimic the on-deck incubators used for the Fe experiments.  The Fe experiment is not part of this study, but influenced sampling strategies throughout the cruise for data comparison and compatibility. The bottom of the euphotic zone ranged between samples between 40-46m. Light depths were determined using the CTD-mounted PAR meter during our daily productivity casts enabling consistent sampling at 46% and 1% light levels.

L250: Any other source(s) of N2O other than photoinhibition?

Good Q, added some text to the methods and results section regarding uncertainty in NCP due to potential mixed layer nitrification, which was small.

L348-358:

"We note that several recent studies have observed nitrification within the euphotic zone, challenging the assumption that $N_2O$ production is limited to subsurface waters (Grundle, Juniper and Giesbrecht, 2013; Smith *et al.*, 2014), and potentially leading to overestimates in our vertical mixing-corrected NCP estimates. Previous observations in the CCS reported a range of depth-integrated mixed layer nitrification rates between $0.3 – 2$ mmol $NH_4^+$ $m^{-2}$ $d^{-1}$, resulting in consumption of $0.6 – 4$ mmol $O_2$ $m^{-2}$ $d^{-1}$ (Stephens *et al.*, 2020). Following the approach of Izett et al. (2018) we used a range of realistic $N_2O:O_2$ stoichiometries to estimate potential upper and lower bounds of mixed layer $N_2O$ production. We determined mixed layer $N_2O$ production likely ranged between $0.09 – 0.23$ $\mu$mol $N_2O$ $m^{-2}$ $d^{-1}$, which would yield offsets in our final NCP estimates between 1.2 and 3.3 mmol $O_2$ $m^{-2}$ $d^{-1}$. Total uncertainty due to sources of error in other derived parameters was determined by following Izett (2021).

L627 – 629:

"Uncertainty in vertical-mixing corrected NCP due potential mixed layer nitrification (see sect. 2.6) represented between $1.5 – 4.2\%$ of our mean corrected NCP value."

L321 and elsewhere: How did you measure salinity? If from CTD, psu is no longer used. Just unitless.

Salinity was measured with a CTD and TSG. Psu units removed.

L354: In general, the Redfield ratio does not apply to surface waters I reckon because Redfield ratio 16 was measured in deep waters. I would rather recommend C: N allocation ratios in algal cells as discussed above.

Agreed that incorporating measurements of C:N in biomass (and Si) would have strengthened the manuscript. Unfortunately, that is not data we have on hand, so our discussion is limited to our measurements of dissolved N, P, and Si.

Figure 7: As you discuss, Cape Bianco had lower temperatures due to stronger upwelling. It is quite interesting but did this temperature difference affect (1) the enzymatic activity of xanthophyll pigment synthesis and consequently NPQ dynamics/xanthophyll cycle (XC)? and (2) transcriptomic regulation such as low-temperature adaptation, which overlayed Fe-regulated gene expression or TPM counts?

Regarding (1), NPQ correlated with SST in Table 1, which might be due to temperature-dependence XC or NPQ? Also, differences in the community structure might influence NPQ or other FRRf parameters?

Yes, NPQ, like Fv/Fm and other photophysiological parameters should vary with temperature and community structure. However, direct temperature effects on NPQ are expected to increase NPQ. Previous lab studies by Xu et al. (2012) have shown that NPQ increases under cold temperatures, while Yan et al. (2019) demonstrated that Arctic phytoplankton enhanced NPQ photoprotective strategies under colder temperatures.

Presumably, slower enzyme activity in the cold reduces capacity for protein repair and downstream electron transport, increasing susceptibility to photodamage or inhibition (Fanesi et al,.2016). However, our data shows a positive correlation between NPQ and SST, opposite of the expected trend due to direct temperature effects. Here, SST acts as an indicator for nutrient-rich upwelling water. While the lower temperatures almost surely have an effect on enzymatic rates, it appears indirect sst effects, and therefore greater nutrient availability, are outweighing direct effects.

Table 1: In the 3rd row, Pmax should be ETRmax??

The Pmax refers to Pmax from photosynthesis-irradiance curves. Pmax is defined by eqn 3 in section 2.3 and the Pmax plotted in Fig 4b. Admittedly, Pmax/Vmax/Cmax terminology all got a bit muddled in section 4.3.1 and is likely the source of confusion here. All of the terminology throughout has been revised for consistency. $P_{max}$ is the maximum rate for ETR. $P_{max-C}$ is the maximum rate of carbon fixation.

Figure 5(c): GP at Cape Bianco is highly variable with numerous spiky up-downs compared with the peaceful Cape Mendocino. Any possible explanations for the difference? Or simply, large errors for estimation of GP in Cape Bianco?

Spiky data around May 31-June 1, indeed! Yet $ETR_{PSII}$, which GP derives from is not spiky around those days. This indicates the variability is coming from the conversion factor from ETR to GP. To convert ETR to GP, we applied the following unit conversion:

$$\frac{e^-}{RCII\ s} * \frac{mmol\ O_2}{4\ mmol\ e^-} * \frac{8.64E4\ s}{d} * \frac{RCII}{400-700\ Chl} * \frac{mmol\ Chl}{m^3} * mld = \frac{mmol\ O_2}{m^2\ d}.$$

Within the conversion factor above, the only variables that are not constant are the mixed layer depth and the mixed layer chlorophyll concentration. The chlorophyll data around Cape Blanco from May 31-June 1 was highly variable (see below) and that variability propagated into our GP estimates. Why was Chl data spikier around Cape Blanco compared to Cape Mendocino? Not sure exactly, but we were transiting across hydrographic fronts between offshore waters and the upwelling plume where a strong bloom was beginning to form.

[Figure]

L451: I wanna know x and y! Please add the exact numbers.

Apologies! Text updated: 641-642: "On average, the bottom depth of the euphotic zone was 14 $\pm$ 12 meters deeper than the bottom depth of the mixed layer"

Figure 6: Quite interesting! I'm curious Stations 2 and 9 showed almost the same conversion factors between surface and subsurface. Any possible explanations? Because of intense vertical mixing at both stations?

It is a bit of a befuddling result – Strong similarities between the subsurface and surface samples would make sense in the context of strong upwelling. However, Station 2 (Cape Blanco) and 9 (Cape Mendocino) were very different from one another. Upwelling, estimated from the NOAA CUTI index, was not the same between the two capes (figure below).

[Figure]

Image caption: Study area colored by the NOAA CUTI upwelling index (1km^2 resolution) on May 31 (day of station 2 sampling) and June 10 (day of station 9 sampling). Star shape indicates our position on the day pictured. Indicating Station 2 sampling coincided with strong upwelling while station 9 sampling coincided with strong downwelling. Speculatively, strong vertical mixing can still explain the strong coherence between subsurface and surface samples at these stations although the prevailing direction of vertical transport differed.

Subsections 4.2 and 4.3 are a bit redundant for me even as an aquatic photosynthesis researcher although your discussion is quite interesting! It would be great if you could shorten these paragraphs.

We have re-edited these sections for brevity. While we hope the writing is more concise, we still kept lots of the fundamental information discussed in these sections since they provide critical context for our results.

Subsection 4.3.2: There is no mention of Fe availability. No need here? It would be more interesting if you could include just a few discussions on Fe availability as your manuscript title suggests.

We have updated L1093-995: In contrast to $\phi_{e:C}/n_{PSII}$, there was no significant differences in NCP:ETR$_{PSII}$ between Cape Blanco, Cape Mendocino, or offshore, suggesting limited effects of nutrient limitation on decoupling between ETR$_{PSII}$ and NCP.

We also added some text that explains our result, per Reviewer 1's suggestion:

L1135-1147: Despite the inherent dependency of net oxygen production on gross oxygen production, the strength of the correlation between NCP and ETR$_{PSII}$ and the consistency of ETR$_{PSII}$:NCP across offshore, Cape Blanco and Cape Mendocino subregions is surprising given the vast suite of potential methodological and physiological sources of uncoupling (Fig 7). However, the derivations of NCP and GP both have similar dependencies on mixed layer Chl concentration. To obtain FRRF-derived GP estimates in comparable units of mmol $O_2$ m$^{-2}$ d$^{-1}$, we multiplied in-situ ETR$_{PSII}$ by mixed layer Chl concentrations (Eq 5). While mixed layer Chl concentrations are not explicitly included in NCP calculations (Sect 2.6), biomass is expected to be a primary driver of bulk productivity. If Chl-normalized NCP is instead compared against GP expressed in units of mmol $O_2$ Chl$^{-1}$ d$^{-1}$, the correlation between 24h binned and instantaneous NCP and ETR$_{PSII}$ estimates decrease to $\rho$ = 0.22 and 0.35, respectively. We therefore conclude that it remains challenging to derive gross and net carbon fluxes from FRRF measurements alone, but paired ETR$_{PSII}$ and Chl measurements can provide useful constraints for NCP estimates.

L685: OK, I would partially agree with your rough estimation using 400-700 chls/RCII. Is it also applicable to Fe-limited waters such as Cape Mednonico? If you look through other chls/RCII ratios in Fe-limited waters (e.g., Strzepek et al. 2019 with appendices and references therein).

Chl:RCII is expected to increase in Fe limited waters. Previously, Schuback et al. (2015) estimated Chl:RCII as 500 for Fe replete water and 700 for Fe limited water based on literature values from Greene et al., 1992.

We have added significant discussion surrounding Chl:RCII (L1009-1033) that will be of interest:

"In addition to non-linear electron transport, $\phi_{e:C}/n_{PSII}$ is also directly affected by the number of Chl energetically coupled to RCII ($1/n_{PSII}$). Directly measuring $1/n_{PSII}$ requires specialized $O_2$ flash yield instrumentation (Suggett *et al.*, 2009), which was unavailable for this study. However, in-situ Chl concentrations, normalized to FRRF-derived proxies for [RCII] ($\propto F_o/\sigma_{PSII}$) following the approach of Oxborough et al. (2012), can be used to examine variability in $1/n_{PSII}$ between samples. With a known instrument calibration factor, $K_a$, either provided by instrument manufacturers or determined independently by $O_2$ flash yield measurements, this approximation could be used to estimate the absolute value of $1/n_{PSII}$.

It is well established that Chl:RCII ($1/n_{PSII}$) ratios increase under low light, to maximize light absorption (Greenbaum and Mauzerall, 1991). In our measurements, the proxy for $1/n_{PSII}$ varied significantly between sample depths, with higher $1/n_{PSII}$ at the bottom of the euphotic zone compared to surface depths, confirmed by a t-test comparison of population means (p << 0.01). Iron limitation is also expected to increase Chl:RCII. Although iron limitation lowers total cellular Chl content, Chl is more likely to be energetically coupled to RCII rather than PSI reaction centers (Greene *et al.*, 1992). Accordingly, $1/n_{PSII}$ displayed a negative correlation with *Fea1* expression in surface samples ($\rho$ = -0.72, p <0.05, n = 9), which we used as a proxy for iron limitation. We thus conclude that Fe-stress likely contributed to variability in $\phi_{e:C}/n_{PSII}$ in addition to influencing non-linear electron transport. The hyperbolic relationship between carbon fixation and electron transport was unaffected by $1/n_{PSII}$, which was assumed to be constant for individual samples throughout the course of photosynthesis-irradiance experiments (Appendix 4). However, this assumption may be violated under high light, due to photoinactivation of RCII (Campbell and Serôdio, 2020). A robust understanding of $\phi_{e:C}/n_{PSII}$ variability requires direct [RCII] measurements collected in parallel with ETR$_{PSII}$ and carbon fixation measurements."

L731: Energy imbalance due to photosystem impairment can be indeed inferred from NPQNSV as you discuss. I would also suggest looking at qP or qL as described above.

Added in details on qP (referred to in our manuscript as F'q/F'm) per your suggestion (Table 1).

---

## Author Response (AR2)

June 12, 2025

Response for reviewers

We wish to thank the reviewer and Associate Editor for their continued efforts to improve this manuscript. Their comments are copied below, followed by our point-by-point responses in red.

**Reviewer 1 Comments**

L176-177: "Light was supplied evenly by the five actinic LEDs within the FRRF (445, 470, 505, 530, and 590 nm)…."

I suppose your FRRf LEDs were surely calibrated but all of which were at the same levels. As you showed in Appendix B, diatoms were exclusively dominant; however, there might be some optical/spectral differences between waters and between surface and subsurface waters. Although it could have minimal effects on $\Phi e:NCP$, is there any possibility that the evenly calibrated LEDs partly decouple $\Phi e:NCP$? Recent FRRf automatically tunes the excitation spectral distribution as the manufacturer says.

All lamps were calibrated against a handheld light meter prior to deployment (L177-178).

The reviewer brings up an important point about spectral corrections. We have added a new section to the discussion (copied below) highlighting the potential effects of spectral differences between instrument light sources and in-situ light environments on the relationship between $ETR_{PSII}$, C-fixation, and NCP.

**4.3.3 Final methodological considerations: spectral corrections**

As discussed above, decoupling between $ETR_{PSII}$, C-fixation and NCP is affected by methodological factors, including differences in the time scale of different photosynthetic processes, and the different normalizations for various measured rates (e.g. per volume, Chl or RCII). Additionally, differences in the spectral characteristics of the in-situ light environment and instrument light sources may contribute further to decoupling between $ETR_{PSII}$, C-fixation, and NCP.

Phytoplankton exhibit variable light use efficiencies across the photosynthetically available wavelength spectrum, due to non-uniform pigment compositions and absorption spectra across assemblages. Consequently, photosynthetic measurements are wavelength-dependent (Kyewalyanga, Platt and Sathyendranath, 1997). As a result, differences in the spectral distribution of light between the FRRF and photosynthetron incubator (used for $^{14}C$ uptake measurements) and the in-situ light environment could influence the stoichiometry we observed between $ETR_{PSII}$, C-fixation, and NCP.

In principal, spectral corrections, can be used to account for variability between instrument light sources and in-situ light environments to improve the inter-comparability of measurements (Schuback., et al., 2021; Tortell et al., 2023). These corrections require

measurements of the spectral distribution of FRRF, photosynthetron incubator, and in-situ light, and a reconstruction of photosynthesis absorption spectra based on photosynthetic pigment concentrations, determined from HPLC analysis. These corrections rely on the assumption that absorption spectra of photosynthetic pigments accurately represent the action spectra of photosynthesis, which not always the case (Kyewalyanga, Platt and Sathyendranath, 1997). In our study, we did not collect measurements of the spectral distribution of light in the euphotic zone, which would have likely varied significantly across depths and sampling sites (i.e. on-shore versus off-shore). For this reason, we are not able to spectrally correct in vitro measurements of $ETR_{PSII}$ and $^{14}C$-uptake to in-situ spectral fields for more direct comparison with NCP. As a result, spectral differences between instrument light sources and the in-situ light environment could contribute to some of the observed variability between $ETR_{PSII}$, C-fixation and NCP in this study. As for the comparison of FRRF and $^{14}C$ data, spectral corrections would affect the absolute magnitude of $\phi_{e:C}/n_{PSII}$, but these corrections would yield a station-specific scalar and would not affect the hyperbolic relationship observed between $ETR_{PSII}$ and C-fixation.

The best approach to minimize the influence of spectral variability is to match the spectral properties of instrument light sources to ambient light fields. However, this remains challenging for high-resolution underway applications across varying spectral environments.

Also is your Φe:NCP prediction successful due to the diatom-dominated community? Or would be successful even in various algal communities? Any possible ideas?

We hypothesized that the strength of the correlation between ETR-derived GP and NCP was primarily due to similar dependencies on [Chl]. See L892 – 925 copied below.

'Despite the inherent dependency of net oxygen production on gross oxygen production, the strength of the correlation between NCP and $ETR_{PSII}$ and the consistency of $ETR_{PSII}$:NCP across subregions is surprising given the multitude of methodological and phsyiological factors that can uncouple these rates (Fig 7). However, the derivations of NCP and GP both have similar dependencies on mixed layer Chl concentration. To obtain FRRF-derived GP estimates in comparable units of mmol $O_2$ $m^{-2}$ $d^{-1}$, we multiplied in-situ $ETR_{PSII}$ by mixed layer Chl concentrations (Eq 5). While mixed layer Chl concentrations are not explicitly included in NCP calculations (Sect 2.6), biomass is expected to be a primary driver of bulk productivity. If Chl-normalized NCP is instead compared against GP expressed in units of mmol $O_2$ $Chl^{-1}$ $d^{-1}$, the correlation between 24h binned and instantaneous NCP and $ETR_{PSII}$ estimates decrease to $\rho$ = 0.22 and 0.35, respectively. We therefore conclude that it remains challenging to derive gross and net carbon fluxes from FRRF measurements alone, but paired $ETR_{PSII}$ and Chl measurements can provide useful constraints for NCP estimates.'

Thank you for providing your transcriptomic data. It looks interesting %diatom was high with high %protists and vice versa. However, I'm curious what the-axis of Fea1 bars indicates. The "counts" indicated "calibrated" counts referenced reads of "diatoms" like

"TPM reference to the total diatom reads"? The Y-axis needs more explanation/clarification,

We have added a few details to the figure caption. Counts were normalized using DESeq2's median of ratios method (Love, Huber and Anders, 2014), and refer to reads mapping to diatoms.

**Editor Comments**
With the next revision, please re-name the appendix figures to B1 and C1.

Figure labels updated.

---

## Author Response (AR3)

July 17, 2025

Response for reviewers

We wish to thank the reviewer and Associate Editor for their attention to detail. Their comments are copied below, followed by our point-by-point responses in red.

First, we have modified our references in accordance with the standards of *Biogeosciences*. For papers with three authors, we have changed the in-text citations to the first author name followed by *et al.* The reference list has been reordered so that co-author papers appear before team papers led by the same first author. Finally, we have added 'a' and 'b' to the year of papers written by the same author in the same year to differentiate between different studies.

**Reviewer 1 Comments**
Thank you for your edits and clarification.

Lastly, I have one technical suggestion for your response regarding Fig. C in your supplementary files.
You mentioned the R-based commands you used (i.e., DEseq2) in your response letter.

However, the caption does not specify that you used Deseq2 and why you used these R plus bioconductor.

I now fully understand what DEseq2 does on your RNA-seq data after reading through "Why un-normalized counts?" at the Deseq2 home page: https://bioconductor.org/packages/devel/bioc/vignettes/DESeq2/inst/doc/DESeq2.html#why-un-normalized-counts

As figure captions need to stand-alone, I suggest you reinforce the description of your strategy, possibly with citations (at least (Love, Huber and Anders, 2014)), why you adopted DEseq2?

Following the reviewer's suggestion, we have added the following statement to our Appendix C caption:

"Normalized *Fea1* counts were calculated using DESeq2's median of ratios method in R (Love, Huber and Anders, 2014)."

Further information regarding our RNA data analysis workflow is available in Methods Section 2.8:

"Raw reads were trimmed using Trim Galore 0.6.10 (Martin, 2011) and quality control was determined with FastQC (Andrews, 2010). A *de novo* metatranscriptome assembly was conducted using rnaSPAdes 3.15.5 (Bushmanova *et al.*, 2019) and CD-HIT-EST (Li and Godzik, 2006). Contigs were annotated using the Marine Functional Eukaryotic Reference Taxa (MarFERReT) database (Groussman *et al.*, 2023), which provides NCBI taxonomic annotations

(Federhen, 2012) and Pfam 34.0 functional annotations (Mistry *et al.*, 2021). Samples were mapped against the MarFERReT DIAMOND sequencing aligner and its compatible BLASTX command (e-value < 1e-06) (Buchfink, Xie and Huson, 2015). Trimmed samples were aligned using Salmon (Patro *et al.*, 2017). The package tximport (Soneson, Love and Robinson, 2016) was used to generate a comprehensive table of read count data for each sample and each contig. Only counts taxonomically mapping to Bacillariophyta (i.e., diatoms) were included. The normalized counts for all genes were then calculated using DESeq2's median of ratios method (Love et al., 2014)." (L365 – 376).

To avoid repetition, we do not reiterate our methods in the Appendix C figure caption.